# Slowing the body slows down time perception

Rose De Kock[1], Weiwei Zhou[1], Wilsaan M Joiner[1], Martin Wiener[2]*

[1]University of California, Davis, Davis, United States; [2]George Mason University, Davis, United States

**Abstract** Interval timing is a fundamental component of action and is susceptible to motor-related temporal distortions. Previous studies have shown that concurrent movement biases temporal estimates, but have primarily considered self-modulated movement only. However, real-world encounters often include situations in which movement is restricted or perturbed by environmental factors. In the following experiments, we introduced viscous movement environments to externally modulate movement and investigated the resulting effects on temporal perception. In two separate tasks, participants timed auditory intervals while moving a robotic arm that randomly applied four levels of viscosity. Results demonstrated that higher viscosity led to shorter perceived durations. Using a drift-diffusion model and a Bayesian observer model, we confirmed these biasing effects arose from perceptual mechanisms, instead of biases in decision making. These findings suggest that environmental perturbations are an important factor in movement-related temporal distortions, and enhance the current understanding of the interactions of motor activity and cognitive processes.

## Introduction

Interval timing is an essential part of survival for organisms living in an environment with rich temporal dynamics. Biologically relevant behaviors often require precise calibration and execution of timed output on multiple levels of organization in the nervous system (*Cisek and Kalaska, 2010*). For example, central pattern generators produce basic locomotion in many organisms and yield balanced, rhythmic motor output via the oscillatory properties of inhibitory interneurons (*Guertin, 2009*). At greater levels of complexity, many behaviors rely on the explicit awareness of time (*Buhusi and Meck, 2005*). Subjective time is not always veridical, however; in fact, across many organisms, it is subject to distortion (*Malapani and Fairhurst, 2002*). As described by *Matthews and Meck, 2016*, temporal distortions can arise from changes in perception, attention, and memory processes, and are proposed to be directly related to the vividness and ease of representation of a timed event. Interestingly, action properties can also influence perceived time. For example, it has been shown that subjective time on the scale of milliseconds to seconds is influenced by movement duration (*Yon et al., 2017*), speed (*Yokosaka et al., 2015*), and direction (*Tomassini and Morrone, 2016*). More specifically, timed events accompanied by arm movements that are short (*Yon et al., 2017*), rapid (*Yokosaka et al., 2015*), or directed toward the body (*Tomassini and Morrone, 2016*) undergo compression.

These studies grant insight into the importance of action in the context of timing, but they are limited by focusing solely on volitional modulation of movement parameters. Often, organisms encounter changes in the environment that dramatically affect the way motor plans are executed. When these perturbations are encountered, organisms use feedback information to update current and future movement plans (*Shadmehr et al., 2010*). In the following experiments, we sought to modulate the parameters of movement distance and speed by introducing changes in the movement environment itself rather than through instruction or task demands. Participants were required

*For correspondence:
mwiener@gmu.edu

Competing interests: The authors declare that no competing interests exist.

to time auditory tone intervals while moving a robotic arm manipulandum through environments with varying degrees of viscosity. This was tested first in a temporal categorization task, then in a temporal reproduction task with a new group of participants. If it is the case that time perception is biased by movement distance then limiting movement by applying viscosity should lead to underestimation of intervals. In our previous work (*Wiener et al., 2019*), we utilized a very similar free-movement categorization paradigm to study the effect of movement on time perception. Unlike the current study, this paradigm had no viscosity factor, but rather tested whether participants that were allowed to move during timed intervals differed in performance from participants that were not allowed to move. Allowance of movement enhanced temporal perception by reducing variability (i.e. lower coefficient of variation). However, results observed in this study were mechanistically ambiguous. That is, we observed that temporal judgments were more precise with movement, but it was unclear whether this effect was driven by perceptual changes or modulation of decision properties (*Figure 1a*). Thus, in the current study, we sought to provide a more mechanistic explanation of our observed results by disentangling perceptual effects and ensuing downstream processes (choice selection in the categorization experiment, and measurement and estimation in the reproduction experiment). In the categorization experiment, we demonstrate that viscosity successfully decreased movement distance, and that this decrease was associated with underestimation of time intervals. We verified that this modulation was a result of interval timing and not a decision-related bias by applying a recently developed drift-diffusion model of timing (*Balcı and Simen, 2014*). In the reproduction experiment, all participants tended to overestimate durations, but viscosity was related to decreased overestimation and greater central tendency. We utilized a Bayesian Observer Model (*Jazayeri and Shadlen, 2010*; *Remington et al., 2018*) to verify that this effect was a result of perceptual bias rather than increased noise in the measurement and production processes. Overall, these results suggest that movement distance has a direct influence on perceived interval length, regardless of whether this parameter is modulated by volitional or environmental factors.

## Results

### Experiment 1 - temporal categorization

In our first experiment, 28 human subjects engaged in an auditory temporal categorization task using supra-second intervals between 1 and 4s. Subjects were required to classify each interval as 'long' or 'short', compared to the running average of all previously experienced intervals. To classify each interval, subjects were required to move the arm of a robotic manipulandum to one of two response locations, counterbalanced between subjects. Prior to tone onset, subjects were allowed a 2s 'warm-up' period, in which they were free to move the cursor around and explore the environment. During this period, the resistive force (f) against the manipulandum was gradually increased to reach a peak viscosity (v) of four possible levels (0, 12, 24, or 36 Ns/m$^2$; see Materials and methods); the viscosity remained at this level for the remainder of the trial (*Figure 1b*). Entry into the response location prior to the tone offset was penalized by restarting the trial, and so the optimal strategy was to move the cursor closer to the 'short' location, and then gradually move to the 'long' location as the tone elapses (*Wiener et al., 2019*). Consistent with this strategy, we found that the relative location of the hand at interval offset was closer to the short target for intervals at or under the middle of the stimulus set, but rapidly moved closer to the long target for longer intervals [$F_{(6,162)}$=4.791, p < 0.001, $\eta^2_p$=0.151] (*Figure 1—figure supplement 1A*). However, no impact of viscosity was observed on relative hand position [$F_{(3,81)}$=1.595, p=0.197], suggesting that movement had little impact on the ability of subjects to employ this strategy. One possible explanation for this lack of an effect is that the introduction of viscosity altered the optimal positional strategy across participants; indeed, external movement perturbation on choice reaching tasks similar to this one reveal that movement strategies change in response to additional effort (*Burk et al., 2014*). We further note that, in this task the position at offset did not guarantee that subjects would choose the closer response location. We further verified that our viscosity manipulation worked by observing a decrease in movement distances [$F_{(3,81)}$=21.05, p < 0.001, $\eta^2_p$ = 0.438] and an increase in force applied to the arm with higher viscosities [$F_{(3,81)}$=22.736, p < 0.001, $\eta^2_p$ = 0.457] (*Figure 1—figure supplement 1A*).

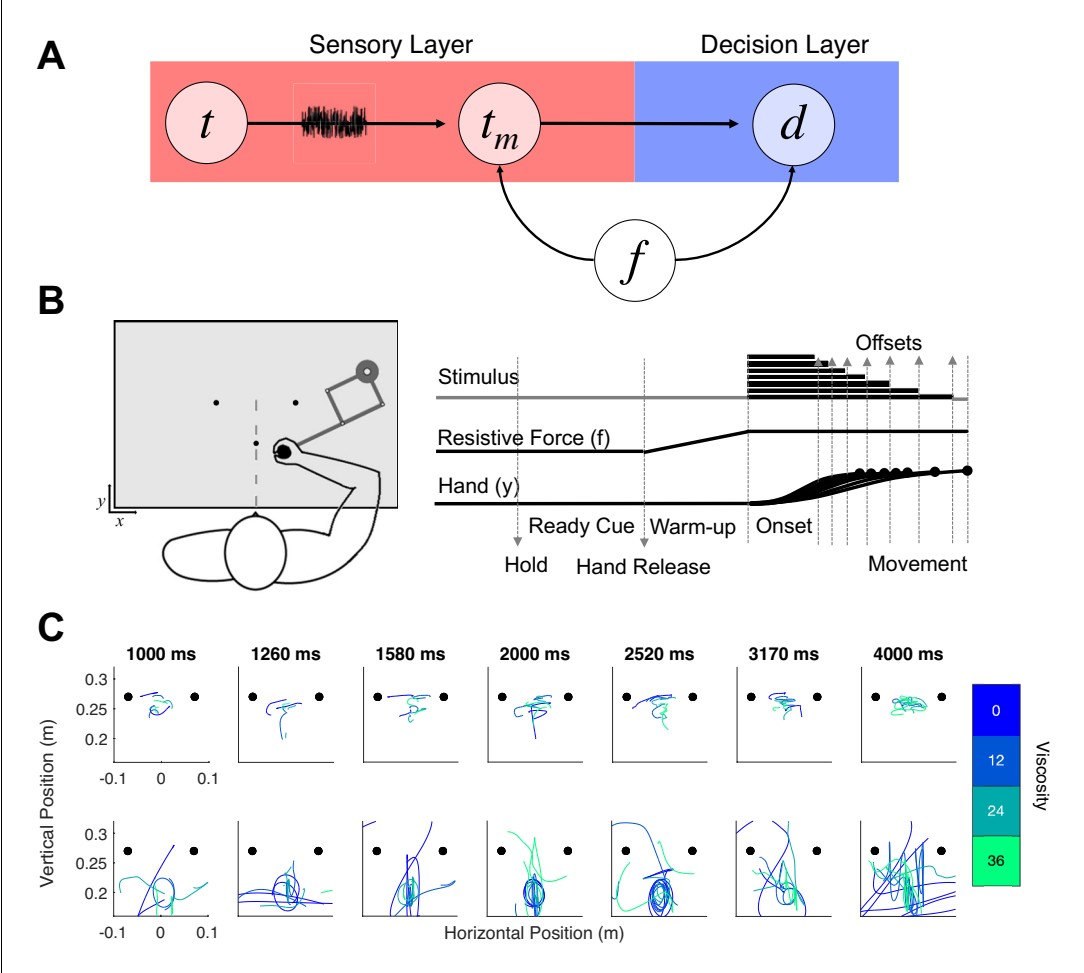

**Figure 1.** Hypothesis and design of Experiment 1. (**A**) Potential pathways in which movement (f) could influence timing. The first possibility is that f specifically alters the sensory layer, in which a stimulus presented for an amount of time (t) is perceived with noise as a temporal estimate ($t_m$); here, f could specifically alter the measurement process, either by shifting the way that estimate is perceived or by altering the level of noise. The second possibility is that f shifts the decision layer, such that decisions about time (d) are biased to one choice or another (e.g. more likely to choose 'long'). (**B**) Task schematic of Experiment 1. Participants began each trial with the robotic handle locked in a centralized location. The trial was initiated by a warm-up phase in which the hold was released and viscosity was applied in a ramping fashion until the target viscosity was reached. Participants were allowed to move throughout the workspace during warm-up and tone presentation, and reach to one of two choice targets to indicate their response (Hand y data shows hypothetical paths to the chosen target). (**C**) Example trajectory data; each row displays sample trajectories from two subjects. The trajectories include movement during the tone for the seven possible tone durations for each of the four viscosities.

The online version of this article includes the following figure supplement(s) for figure 1:

**Figure supplement 1.** Additional effects for categorization and reproduction tasks.
**Figure supplement 2.** Individual differences in movement parameters for categorization and reproduction tasks.

Analysis of choice responses proceeded by constructing psychometric curves from the mean proportion of 'long' response choices for each interval/viscosity combination, and chronometric curves from the mean reaction time (RT) as well (*Figure 2a* and *Figure 1—figure supplement 1*). Psychometric curves were additionally fit with cumulative Gumbel distributions, from which the bisection point (BP) was determined as the 0.5 probability of classifying an interval as long. Analysis of the BP values across all four viscosities with a repeated-measures ANOVA revealed a significant effect of viscosity [$F(3,81)=3.774$, $p=0.014$, $\eta^2_p = 0.123$]. A further examination revealed this to be a linear effect, with BP values generally increasing with viscosity, indicating a greater tendency to classify intervals as 'short' [$F(1,27)=5.439$, $p = 0.027$, $\eta^2_p = 0.168$]; we further note that examination of a quadratic contrast did not reveal a significant effect [$F(1,27)=1.209$, $p=0.281$]. To further confirm this effect, we calculated slope values of a simple linear regression of the BP against viscosity across

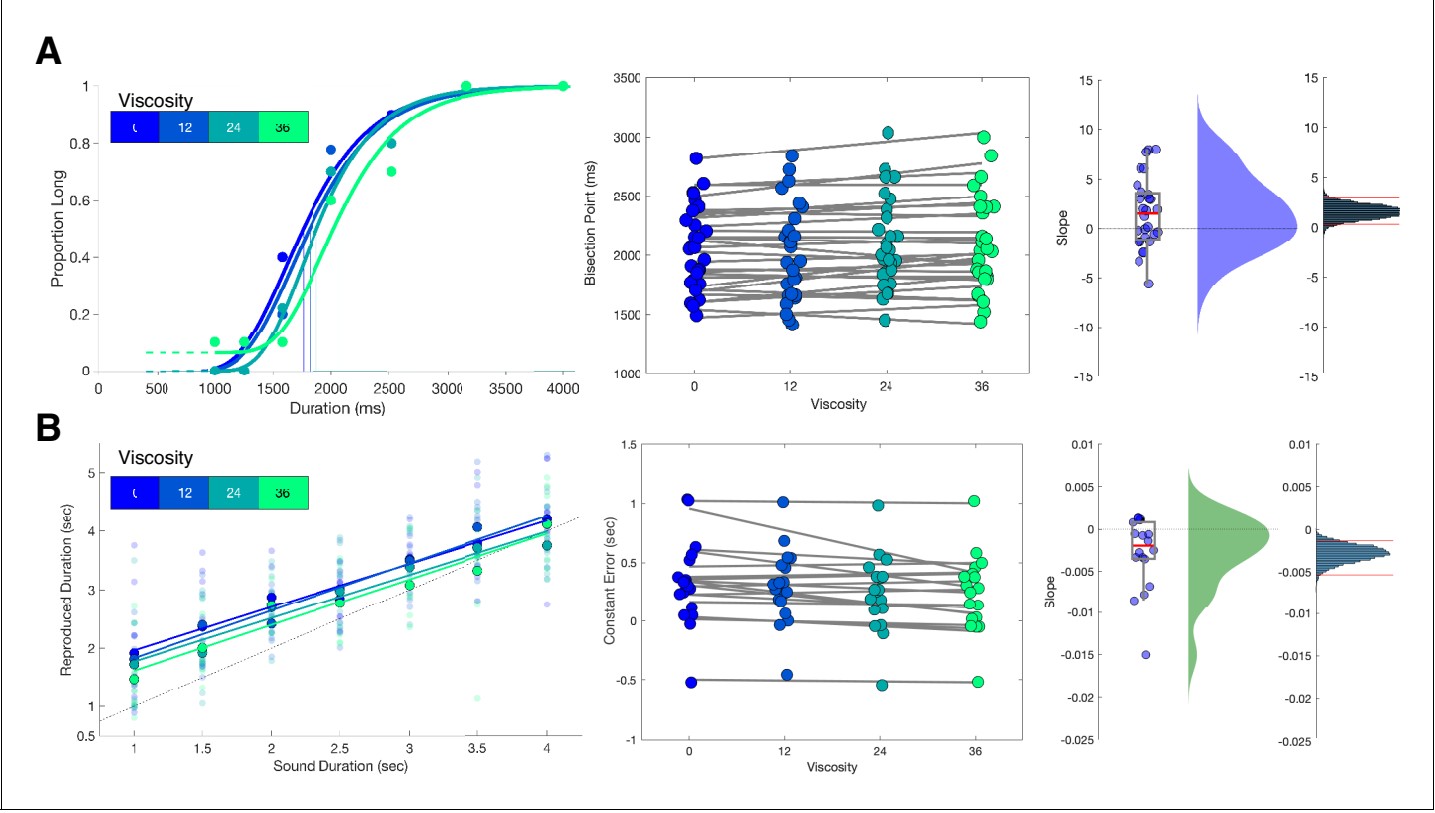

**Figure 2.** Viscosity shifts time responses. (**A**) Results from Experiment 1 (Temporal Categorization). Left panel: psychometric curves fit to response proportions for a representative subject exhibiting a rightward shift with increasing viscosity; vertical lines indicate the Bisection Point (0.5 probability of classifying 'long'). Middle panel: Bisection points for all subjects across viscosities; gray lines represent best fitting linear regressions. Right panels: boxplots and kernel densities of slope values for linear regressions, along with bootstrapped distributions of the mean slope (rightmost panel) with 95% confidence intervals. (**B**) Results from Experiment 2 (Temporal Reproduction). Left panel: Reproduction performance for a representative subject exhibiting progressively shorter reproduced time estimates with higher viscosities; faded points represent single trials, solid points represent means, lines represent best fitting linear regressions. Middle panel: Mean Constant Error (difference between reproduced and presented interval) for all subjects across viscosities; gray lines represent best fitting linear regressions. Right panels: boxplots and kernel densities of slope values for linear regressions, along with bootstrapped distributions of the mean slope (rightmost panel) with 95% confidence intervals.

subjects; a non-parametric bootstrap (10,000 samples) of 95% confidence intervals demonstrated that slope values did not overlap with zero [0.3183 - 2.8563], indicating robustness of the effect.

Analysis of RT values demonstrated faster RTs with longer perceived duration [$F(6,162)=38.302$, $p < 0.001$, $\eta^2_p = 0.587$], consistent with previous reports (*Balcı and Simen, 2014*; *Wiener and Thompson, 2015*; *Wiener et al., 2019*). This pattern is thought to reflect increased decision certainty associated with longer intervals; once an elapsed interval crosses the categorical boundary, subjects shift from preparing a 'short' choice to a 'long' choice, with increased preparation for longer durations. Additionally, a significant effect of viscosity was observed [$F(3,81)=14.684$, $p < 0.001$, $\eta^2_p = 0.352$]; however, the effect was variable across viscosities, with faster RTs for mid-range viscosities. No effect of viscosity was observed on the CV [$F(3,81)=0.377$, $p=0.77$; $BF_{10}=0.073$] (*Figure 1—figure supplement 1*).

## Influence of movement parameters

We further examined the impact of individual differences in movement parameters on the observed behavioral findings. As noted above, movement distance was successfully manipulated by imposing different environmental viscosities. However, we observed large inter-individual differences in the average distances moved by different subjects. That is, some subjects moved a lot, whereas some moved very little; notably, subjects were largely consistent in their movement distances across

viscosity conditions (*Figure 1—figure supplement 2A*). Similarly, subjects who moved more also exerted more force in doing so; we further observed that the effect of viscosity on movement distance and force were correlated between subjects [Pearson r = −0.73, p<0.001; Spearman r = −0.88, p<0.001]. We further examined if the effect of viscosity on time perception was modified by individual differences in movement distance. Here, we found only a weak correlation between the effect of viscosity on movement distance and the bisection point [Pearson r = −0.286, p=0.1395; Spearman r = −0.35, p=0.067]. Similarly, the correlation between the effect of viscosity on force was also very weak [Pearson r = 0.147, p=0.45; Spearman r = 0.221, p=0.25], suggesting that the effect of viscosity did not covary with individual differences in movement distance or force.

While the above results suggested no between-subject difference in the magnitude of the effect, this does not preclude a within-subject influence. In our previous report (*Wiener et al., 2019*), we observed that subjects performing this task exhibited a more precise perception of time (lower CV) compared to subjects who performed a different version where the robotic arm was fixed at the starting point for the duration of the interval. Although the present study allowed all subjects to move freely during the interval, we hypothesized that movement during the interval would interact with precision within-subject. To test this possibility, we performed a within-subject median split of the movement distance for each interval/viscosity combination and re-analyzed the psychometric curves for each viscosity condition. For the BP, we again observed an increase with viscosity [$F(1,27)$ =5.936, p = 0.022, $\eta^2_p$=0.18], regardless of how much subjects moved [$F(1,27)$=1.397, p=0.247] (*Figure 3a*). For the CV, a significant interaction between viscosity and movement distance was observed [$F(1,27)$=7.694, p=0.01, $\eta^2_p$=0.222], in which the CV was significantly lower when subjects moved more, but only when the viscosity was zero [$t(27)$=-2.237, p=0.034, D=1.2] (*Figure 3b*), and not for any other viscosity (all p>0.05). This indicates that precision again improved with greater movement, but only when no impediments from a viscous movement environment existed.

## Drift diffusion modeling

The results of this Experiment appeared to support the hypothesis that increasing viscosity while judging an auditory interval led to a shorter perception of that interval. However, as stated in the Introduction, a shift resulting from increased viscosity could have either altered perception or biased subjects to classify intervals as 'short'; both outcomes could explain our results, as our task inherently involves a directional judgment (*Yates et al., 2012*; *Schneider and Komlos, 2008*). To further tease apart these two possibilities, we decomposed choice and RT data using a drift diffusion model

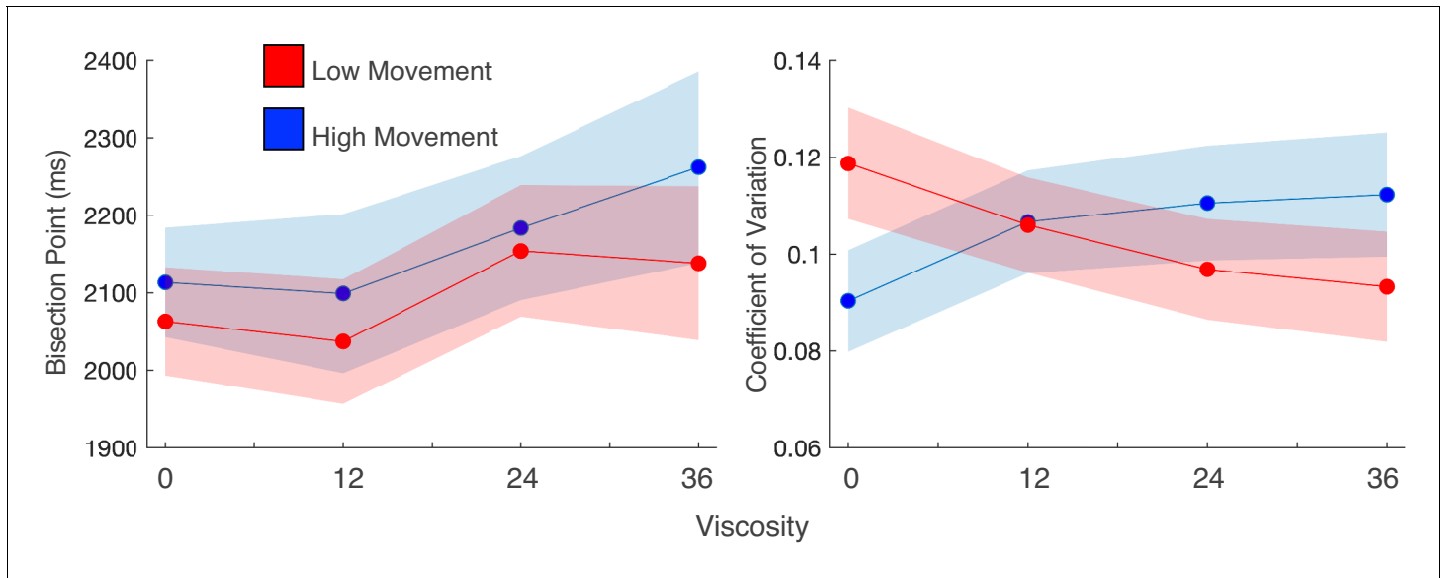

**Figure 3.** Movement speed does not change viscosity effects, but does influence precision. (a) Mean bisection points and (b) coefficients of variation for participant trials divided into high and low movement via a median split. Viscosity shifted the bisection point across both movement types (left); however, precision was influenced by movement, but only when no viscosity existed.

(DDM) of perception and decision making. We employed hierarchical DDM (HDDM; *Wiecki et al., 2013*) in order to constrain fitted parameters for individual subjects by the group mean (see Materials and methods). Under this framework, evidence is accumulated over time towards a 'long' or 'short' decision boundary. A shift in the drift rate towards one of these boundaries is interpreted as evidence in favor of a shift in the perceptual evidence, whereas a change in the threshold boundary could be interpreted as a change in the decision layer (*Voss et al., 2004*; *Hagura et al., 2017*; *Bogacz et al., 2006*).

In constructing our model, we relied on a previously-formulated DDM for describing behavior in temporal categorization tasks (*Balcı and Simen, 2014*). Under this framework, categorization is described as a two-stage process, in which elapsed time is measured during an initial processing stage, which then sets the parameters for a second-stage decision process. Specifically, the first-stage is conceived as a single drift process with a variable drift rate, akin to pacemaker-accumulator models of time perception (*Allman et al., 2014*). At interval offset, the second-stage process begins, in which another drift process is engaged towards either an upper or lower decision boundary for classifying intervals as 'long' or 'short' (*Wiener et al., 2018*). Critically, the starting point (z) and drift rate (v) of the second stage process are determined by the accumulated value at the end of the first-stage process; shorter or longer intervals in the stimulus set lead to starting points closer to the short or long boundaries, respectively and faster drift rates. In this way, the starting point and drift rate are linked, with both parameters set by the perception of the timed interval.

However, the low trial count in our study precluded us from building a DDM that could simultaneously account for both increasing duration and viscosity. To address this, and ensure that our findings accorded with the predictions of the *Balcı and Simen, 2014* model, we constructed two separate DDMs, one for each factor, termed the Duration Model and Viscosity Model (see Materials and methods). For each model, comparisons of model complexity were conducted by comparing DIC values, as well as posterior predictive checks. For the Duration Model, we initially observed that a model which included all four DDM parameters varying with duration (v,a,t,z) was best able to account for subject data [Full model DIC = 6967.01; v,a,t model DIC = 7350.09; v,a model DIC = 8076.33; v model DIC = 9514.65; empty model DIC = 15677.96]. Posterior predictive checks of the data additionally demonstrated that the Duration Model was able to produce a similar pattern to subject data for both choice and RT (*Figure 4—figure supplement 2A*). For individual parameters, we additionally observed patterns that matched those reported by Balci and Simen (2014; see also *Wiener et al., 2018* for a replication of these patterns). These included a change in drift rate from the short to long duration boundary with increasing duration length and a linear increase in the starting point from the short to long. Additional patterns from Balci and Simen (2014) were also observed, including U-shape and inverted U-shape patterns for threshold and non-decision time, respectively (*Figure 4—figure supplement 2B*). Finally, we observed the predicted linkage between starting point and drift rate, with a strong correlation between them [Pearson r = 0.664, p<0.001; Spearman r = 0.618, p<0.001] (*Figure 4—figure supplement 2C*).

For the Viscosity Model, we initially observed that having all four parameters vary by viscosity was not warranted; critically, this was driven by inclusion of the starting point, suggesting that changes in the starting point with viscosity did not improve the model fit [Full model DIC = 15245.36; v,a,t model DIC = 15213.62; v,a model DIC = 15747.6833; v model DIC = 15792.24; empty model DIC = 15678]. However, given our a-priori assumption that all four parameters could vary, and the inclusion of the starting point in the *Balcı and Simen, 2014* model, we report results here for the full model, noting the inclusion of the starting point does not change these findings. In our analysis of the fitted DDM parameters for the Viscosity Model, we observed first a significant shift in the drift rate (v) with increases in viscosity [$F_{(3,81)}=4.562$, p=0.005, $\eta^2_p = 0.145$]; this shift was notably linear in nature, with the drift rate shifting to the 'short' duration boundary with higher viscosities [$F_{(1,27)}=7.866$, p = 0.009, $\eta^2_p = 0.226$]. Further analyses also revealed significant effects of viscosity on the threshold (a) [$F_{(3,81)}=12.356$, p <0.001, $\eta^2_p = 0.314$] and starting point (z) [$F_{(3,81)}=43.73$, p <0.001, $\eta^2_p = 0.618$], but no effect on the non-decision time [$F_{(3,81)}=2.257$, p = 0.088] (*Figure 4b*). In both cases for the threshold and starting point, the dominant pattern was for these values to drop for viscosities above zero, but show little variation beyond that. We additionally note that this pattern can explain the RT findings observed with viscosity; specifically, a lower threshold should be associated with faster RTs. With higher viscosity, and so higher effort, subjects may have placed greater emphasis on speed for their responses, and so lowered the necessary threshold for evidence (*Burk et al., 2014*;

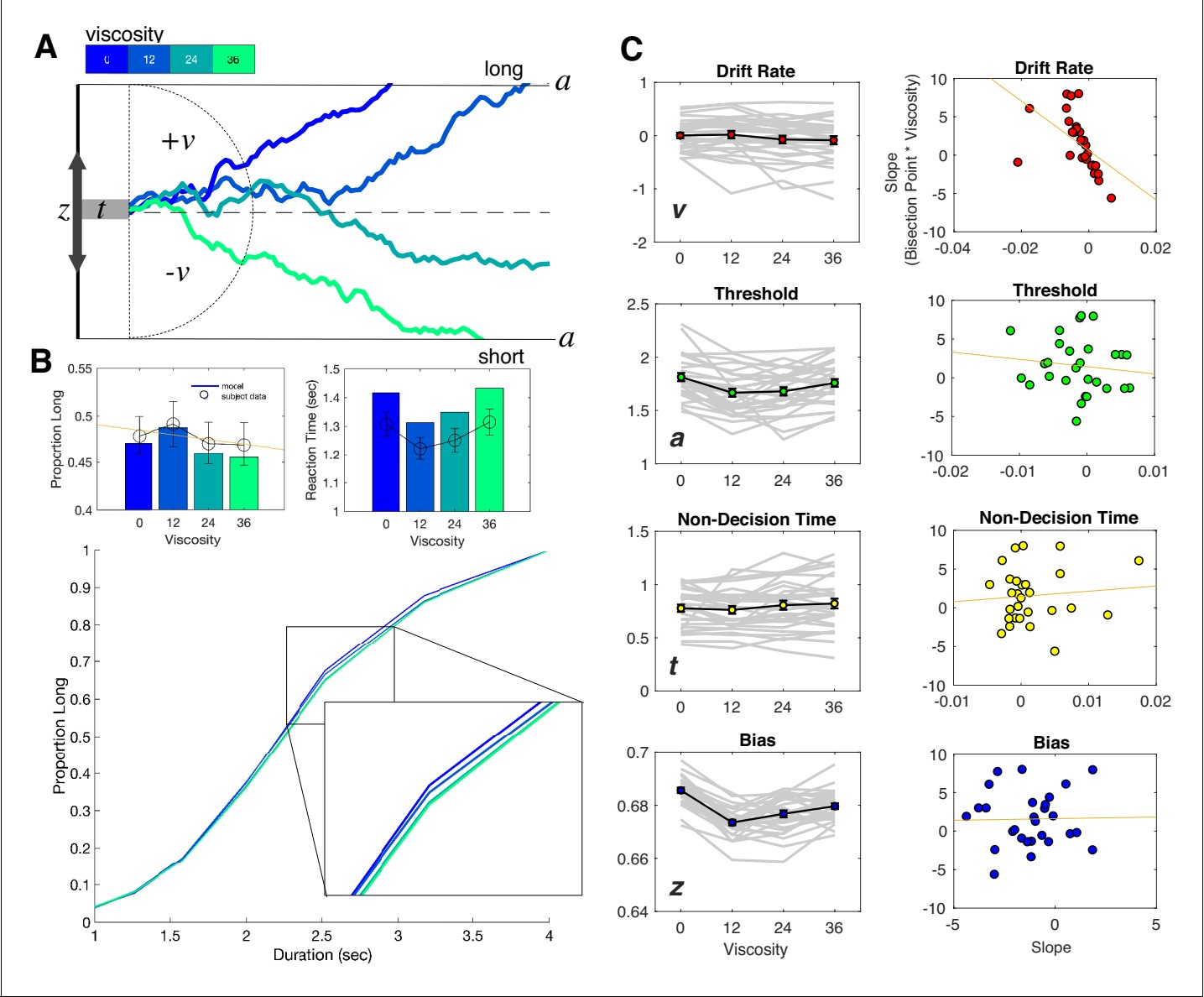

**Figure 4.** Drift diffusion modeling of categorization performance and viscosity. (A) Example Viscosity DDM model, in which evidence is accumulated to one of two decision bounds ('long' and 'short'), separated by a. Evidence accumulation drifts at particular rate (v) that can be positive or negative, depending on the direction of the drift to a particular boundary. The drift rate is additionally delayed by non-decision time (t) and may be biased toward one of the boundaries by a certain amount (z). Viscosity was specifically found to influence the drift rate, in which higher viscosities were associated with a shift in drift from the long to short decision boundary (presented traces represent example simulations). (B) Top Panels: Posterior predictive checks for the Viscosity Model, displaying simulated data (bars) against average subject data for choice (left) and reaction time (right). Bottom Panel: Psychometric curves from simulations of the 'Full' Model, combining Viscosity and Duration; inset displays a shift in Viscosity in choosing 'long' (C) Fitted Viscosity Model results for all four parameters (left panels), showing that viscosity linearly shifted the drift rate, but also modulated threshold and bias parameters in a nonlinear (stepwise) manner. Right panels demonstrate the correlation between the slope of the viscosity effect on each parameter and the slope of the viscosity effect on behavior; only drift rate exhibited a significant correlation (see also *Table 1* for Fisher Z comparisons between correlations).

The online version of this article includes the following figure supplement(s) for figure 4:

**Figure supplement 1.** Comparison of hierarchical and non-hierarchical fits for Experiment 1.
**Figure supplement 2.** Parameters and simulations of the Duration Model for Experiment 1.

*Hagura et al., 2017*). However, we note that this effect dropped off with higher viscosities, further suggesting that the greater effort needed to employ this strategy reduced its effectiveness.

Given the linear pattern observed for changes in the drift rate, we further explored whether this parameter could exclusively explain the shift in the BP. To test this, we calculated the slope of a linear regression for each parameter against viscosity for each subject, and correlated these with the slope values for BP against viscosity. Here, the only significant correlation observed was for the drift rate [Pearson r = −0.5132, p=0.0052; Spearman r = −0.7865, p<0.001], and not for any other parameter (all p>0.05). A Fisher's Z-test comparing this correlation confirmed that it was significantly greater than for all other parameters (see *Table 1*). We additionally confirmed this correlation was correct when using a non-hierarchical method for fitting the Viscosity Model, to confirm that the results were not driven by potential shrinkage resulting from the hierarchical method (*Figure 4—figure supplement 1*; *Katahira, 2016*).

## Experiment 2 - temporal reproduction

The results of Experiment 1 demonstrated that increased resistive force while subjects made temporal judgments about auditory durations led to shorter reported lengths of those durations Computational modeling using a DDM further suggested that this shift was due to viscosity altering the perceived duration, rather than altering decision bias. However, in this experiment, decision-making and perception are intertwined, such that subjects must simultaneously measure the interval duration while classifying it. Indeed, previous research has suggested that, once the categorical boundary (here, the BP) has been crossed, subjects may stop accumulating temporal information altogether (*Wiener and Thompson, 2015*).

To further disentangle whether viscosity impacts perception or decision layers, we had a new set of subjects (n = 18) perform a temporal reproduction task, in which they moved the robotic arm while listening to auditory tone intervals and encoding their duration (*Figure 5*). As a critical difference from Experiment 1, in Experiment 2 subjects were required to move throughout the interval – any halts in movement were penalized by re-starting the trial. This was done following our observation in Experiment one that some subjects chose to move very little. Following the encoding phase, the arm was locked in place and subjects reproduced the duration via a button-press attached to the handle (see Materials and methods). Viscosity was again randomized across the same four levels during the encoding phase; in this way, the impact of resistive force was applied only while subjects were actively perceiving duration, without any deliberative process.

We initially confirmed again that our viscosity manipulation was effective, with reduced movement distance [F(3,51)=149.82, p < 0.001, $\eta^2_p$ = 0.898] and increased force [F(3,51)=114.84, p < 0.001, $\eta^2_p$ = 0.871] observed with greater viscosities (*Figure 1—figure supplement 1B*). For behavioral results, we initially measured the reproduced durations ($t_p$), finding both a main effect of viscosity [F(3,51)=5.5, p = 0.002, $\eta^2_p$ = 0.244] and an interaction with the presented duration [F(3,51) =1.814, p = 0.023, $\eta^2_p$ = 0.096]. No impact on the variance of reproduced estimates was observed, with the CV remaining stable across all viscosities [F(3,51)=0.691, p = 0.562; $BF_{10}$=0.016] (*Figure 1— figure supplement 1B*). More specifically, we observed that reproduced durations generally were overestimated compared to the presented sample durations ($t_s$), and this effect was quantified by measuring the offset for each reproduced duration compared to the presented one, also known as the Constant Error; here, we additionally observed an effect of viscosity, with less overestimation

**Table 1.** Correlation coefficients and Fisher Z comparisons between fitted parameters and behavioral effects.

| | Experiment 1 - Correlation with Viscosity Effect | | | | | Experiment 2 - Correlation with Viscosity Effect | | |
| --- | --- | --- | --- | --- | --- | --- | --- | --- |
| | drift (v) | threshold (a) | starting point (z) | non decision time (t) | | offset (b) | production (p) | measurement (m) |
| Pearson | *0.5132 | 0.1211 | 0.0196 | 0.0926 | Pearson | *0.7332 | 0.0626 | 0.0509 |
| Spearman | *0.7865 | 0.168 | 0.0252 | 0.0733 | Spearman | *0.709 | -0.1022 | -0.0299 |
| Fisher Z compare Pearson with drift | | -3.491 | -1.588 | -2.424 | Fisher Z compare Pearson with drift | | 2.508 | 2.545 |
| Fisher Z compare Spearman with drift | | -5.312 | -3.242 | -3.886 | Fisher Z compare Spearman with drift | | 2.711 | 2.263 |

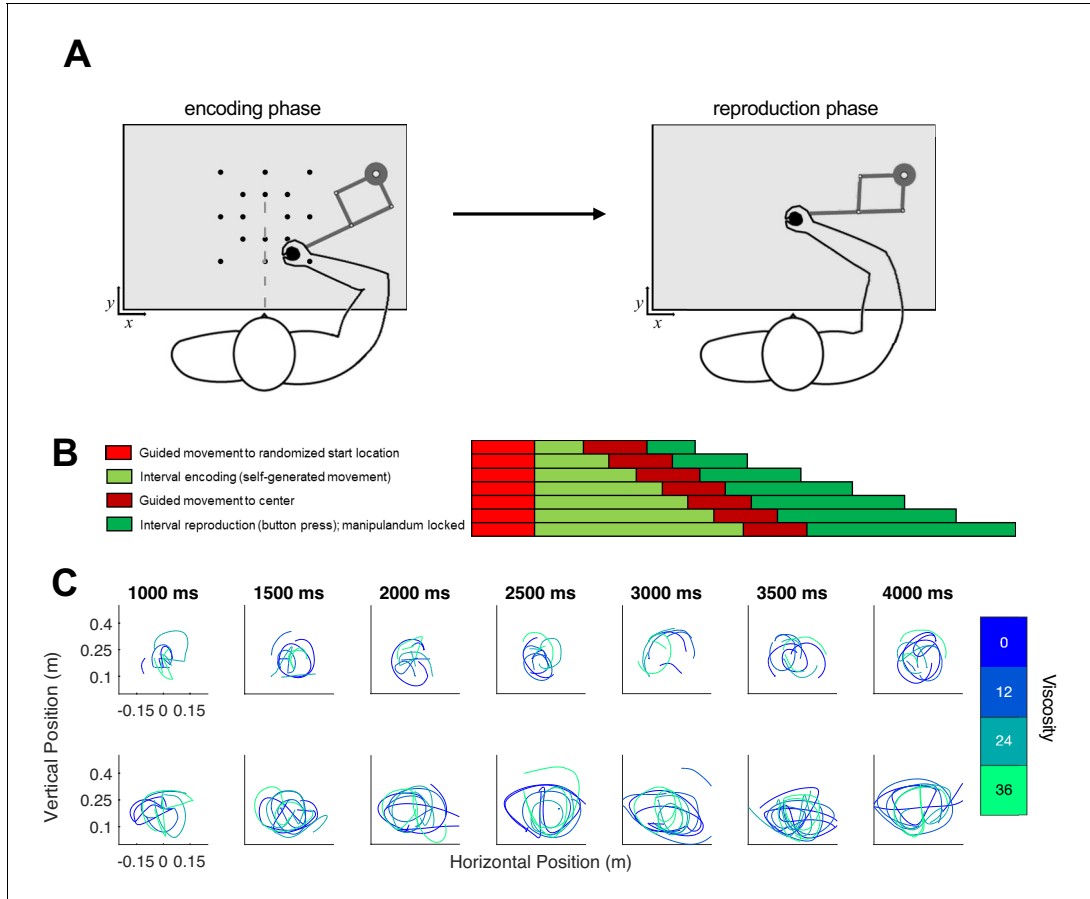

**Figure 5.** Task schematic of Experiment 2. (**A**) Participants began each trial at a randomized start location and were required to initiate movement in order for the test duration to play (encoding phase). Unlike Experiment 1, the desired viscosity was applied immediately rather than in a ramping fashion. Then, the handle was brought to a central location where participants reproduced the duration by holding and releasing a button attached to the handle. (**B**) Timeline for each of the seven tested intervals. (**C**) As in Experiment 1, each row displays sample trajectories from two subjects for the seven possible tone durations, with separate lines indicating different viscosities.

with increasing viscosity [F(3,51)=5.5, p = 0.002, $\eta^2_p$ = 0.244] (*Figure 2B*). We additionally observed an increase in the so-called central tendency effect, in which reproduced durations gravitate to the mean of the stimulus set, with greater viscosities; this effect was quantified by a change in slope values of a simple linear regression [F(3,51)=3.473, p = 0.023, $\eta^2_p$ = 0.17].

## Influence of movement parameters

Similar to Experiment 1, we examined the potential influence of individual differences in movement parameters on the experimental findings. Unlike Experiment 1, subjects were required to continue moving at all times during the interval, and so we predicted less heterogeneity in subject performance. As expected, we observed a close link between movement distance and force exerted, yet with a narrower range for each than in Experiment 1 (*Figure 1—figure supplement 1B*). Unlike Experiment 1, we found no correlation between the effects of viscosity on Movement Distance and Force [Pearson r = −0.16, p=0.52; Spearman r = −0.14, p=0.55], suggesting the correlation observed in Experiment 1 was primarily driven by some subjects moving very little. Additionally, we observed no between-subject correlation between the effects of viscosity on movement distance and duration reproduction [Pearson r = 0.07, p=0.75; Spearman r = 0.19, p=0.44], nor on force and duration reproduction [Pearson r = 0.008, p=0.97; Spearman r = −0.23, p=0.34] (*Figure 1—figure supplement 1B*).

## Bayesian observer model

The results of Experiment 2 revealed that, with increasing viscosity while encoding a time interval, the reproduced interval was increasingly, relatively, shorter in length. Again, this finding is consistent with reduced movement altering the perception of temporal intervals. We note that the temporal reproduction task as designed does not share the overlap with decision-making as in the temporal categorization task, as viscosity was only manipulated while subjects estimated the interval, and was not included during reproduction. However, we also note that the behavioral data alone are somewhat ambiguous to how viscosity impacts time estimation, as we observed both a shift in time intervals, as well as an increase in central tendency with greater viscosities. Changes in central tendency may be ascribed to a shift in uncertainty while estimating intervals, and although the CV did not change across viscosities, it remains possible that viscosity led to greater uncertainty, which would explain the observed shifts.

To tease these two possibilities apart, we employed a Bayesian Observer-Actor Model previously described by Remington and colleagues (*Remington et al., 2018*; *Jazayeri and Shadlen, 2010*) (see Materials and methods). In this model, sample durations ($t_s$) are inferred as draws from noisy measurement distributions ($t_m$) that scale in width according to the length of the presented interval. These measurements, when perceived, may be offset from veridical estimates as a result of perceptual bias or other outside forces (b). Due to the noise in the measurement process, the brain combines the perceived measurement with the prior distribution of presented intervals in a statistically optimal manner to produce a posterior estimate of time. The mean of the posterior distribution is then, in turn, used to guide the reproduced interval ($t_p$), corrupted by production noise (p) (*Figure 6a*). The resulting fits to this model thus produce an estimate of the measurement noise (m), the production noise (p), and the offset shift in perceived duration (b). Note that the offset term is also similar to that employed for other reproduction tasks as a shift parameter (*Petzschner and Glasauer, 2011*).

The result of the model fitting first demonstrated a significant effect on the width of the production noise (p) [$F(3,51)=3.548$, $p = 0.021$, $\eta^2_p = 0.173$] (*Figure 6b*). More specifically, production noise was found to decrease with higher viscosities; however, this effect was not linear, with the only difference being for zero viscosity estimates higher than all others. We note that this effect is similar in form to the shift in the threshold parameter (a) from the Viscosity-DDM of Experiment 1, and so may reflect a change in strategy from higher viscosity. That is, in response to the greater effort during measurement, subjects attempt to compensate by increasing motor precision during production (*Remington et al., 2018*).

For the offset shift (b), we observed a significant effect of viscosity [$F(3,51)=3.72$, $p = 0.017$, $\eta^2_p = 0.18$] that was linear in nature, with a reduction in values with increasing viscosity. No effect of viscosity was observed on the measurement noise parameter (m) [$F(3,51)=1.212$, $p = 0.315$]. As with the DDM results of Experiment 1, we further explored whether the linear nature of the shift in *b* could best explain the observed underestimation of duration by calculating the slope of a regression line for each parameter against viscosity and compared that to the change in reproduced duration. Only the offset term significantly correlated with the underestimation effect [Pearson = 0.7332, $p < 0.001$; Spearman = 0.709, $p<0.001$]. Again, a Fisher's Z-test comparing this correlation confirmed that it was significantly greater than for all other parameters (see *Table 1*).

In order to extend the modeling results further, we sought to compare these findings to alternative version of the Bayesian model. We therefore constructed a second model in which the offset term (b) was moved from occurring at the measurement stage to the production stage (*Figure 7A*). This second model, termed the Viscosity Production Model, was fit to subject data and compared to the first model, termed the Viscosity Perception Model. For comparison, we conducted predictive checks by simulating data from both models and plotting these simulations against the observed subject data (*Figure 7B*). Here, we observed that while the Viscosity Perception Model provided a good fit and description of the data, including a replication of the linear effect of viscosity on constant error, the Viscosity Production Model failed to do so. This observation was confirmed by comparing Negative Log-Likelihood estimates of model fits across subjects and viscosities, in which the Perception Model provided a significantly better fit [$F(1,17)=21.686$, $p<0.001$, $\eta^2_p = 0.561$] (*Figure 7*). We further note that, for the Production Model, the effect of viscosity was not significant [$F(3,51)=0.871$, $p=0.462$], suggesting that the model was not simply shifted from the true response.

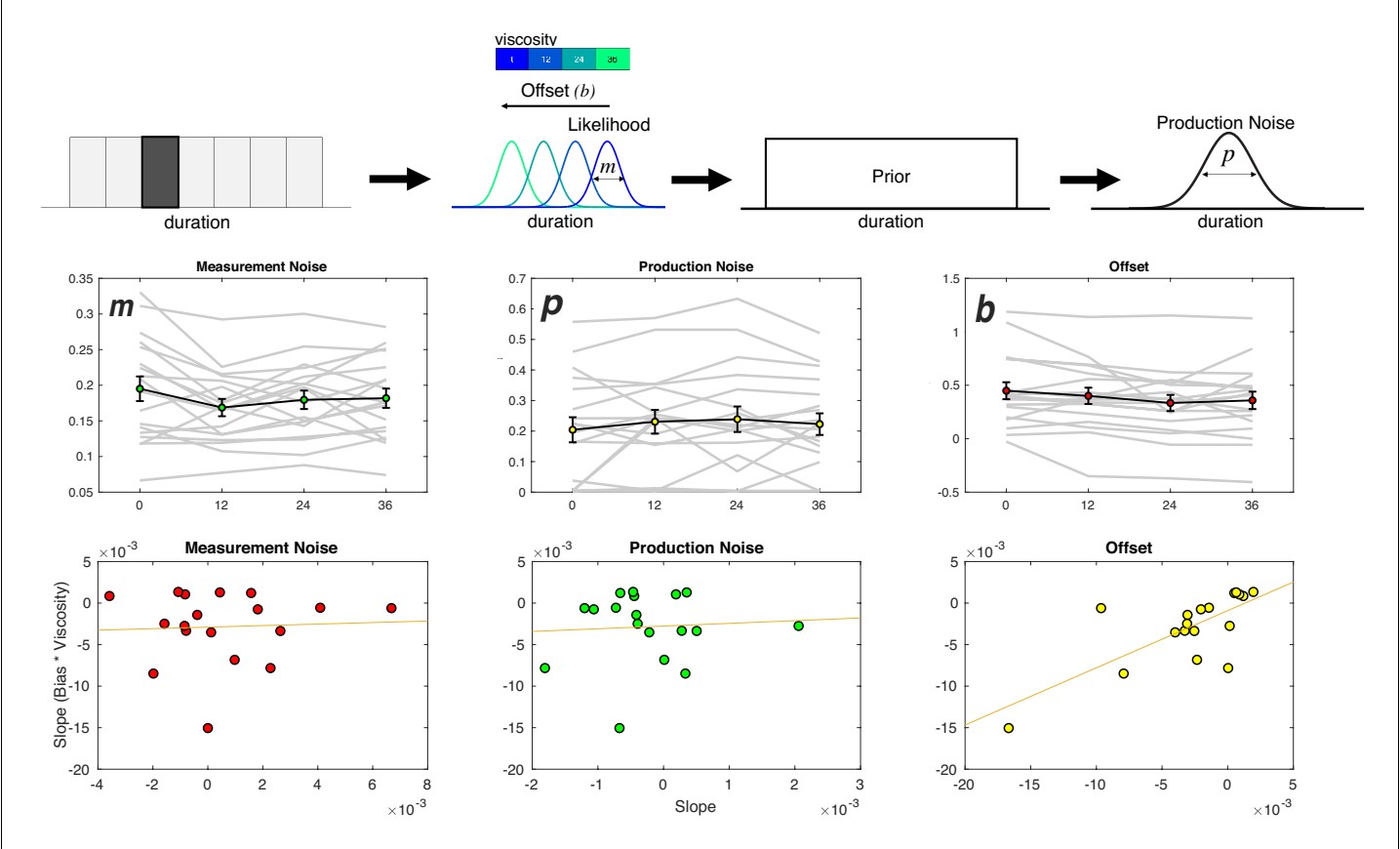

**Figure 6.** Viscosity shifts time reproduction. Top: Bayesian Observer Model. On a given trial, a presented duration is drawn from a likelihood distribution with scalar variance leading to a measurement estimate (m) that is shifted by an offset parameter (b). The measurement estimate is combined with a uniform prior distribution of presented durations, and then finally affected by production noise (p). Viscosity was found to specifically shift b in a linear manner, with greater viscosities associated with shorter perceived durations. Middle panels: Fitted results for all three parameters, demonstrating a linear effect of offset, no effect of measurement noise, and a nonlinear (stepwise) shift in production noise with greater viscosities. Bottom panels display correlations with the behavioral effect of viscosity; only the offset parameters exhibited a significant effect (see *Table 1* for Fisher Z comparisons). Right panel was additionally significant after outlier removal.

Notably, the Production Model was still able to capture the effect of production noise we observed in the Perception Model [$F(1,17)=3.211$, $p=0.031$, $\eta^2_P = 0.159$].

## Discussion

The above experiments demonstrate that systematically impeding movement during interval timing leads to a subsequent compression of perceived duration. These findings complement previous work showing that time perception is highly sensitive to movement (*Yon et al., 2017*; *Yokosaka et al., 2015*; *Tomassini and Morrone, 2016*), and here we confirm a case in which movement parameters (e.g., length and duration) did not have to be self-modulated to induce these distortions. The movement restrictions we implemented (i.e. moving in environments with different manipulations of viscosity) tended to shift the BP later in time in a temporal categorization task, and subsequently shortened perceived intervals in a temporal reproduction task.

In the temporal categorization task, we found that increased viscosity, on average, shifted the BP such that subjects responded 'long' less often. We then applied a drift-diffusion model to isolate the cognitive mechanisms contributing to this effect (i.e. whether it was a function of decision bias, speed-accuracy trade-off calibration, non-decision time, or the rate of evidence accumulation; *Ratcliff, 1978*). The only significant contributor was the drift rate parameter, which linearly shifted from the 'long' to the 'short' boundary with increasing viscosity. While this was evidence for a purely

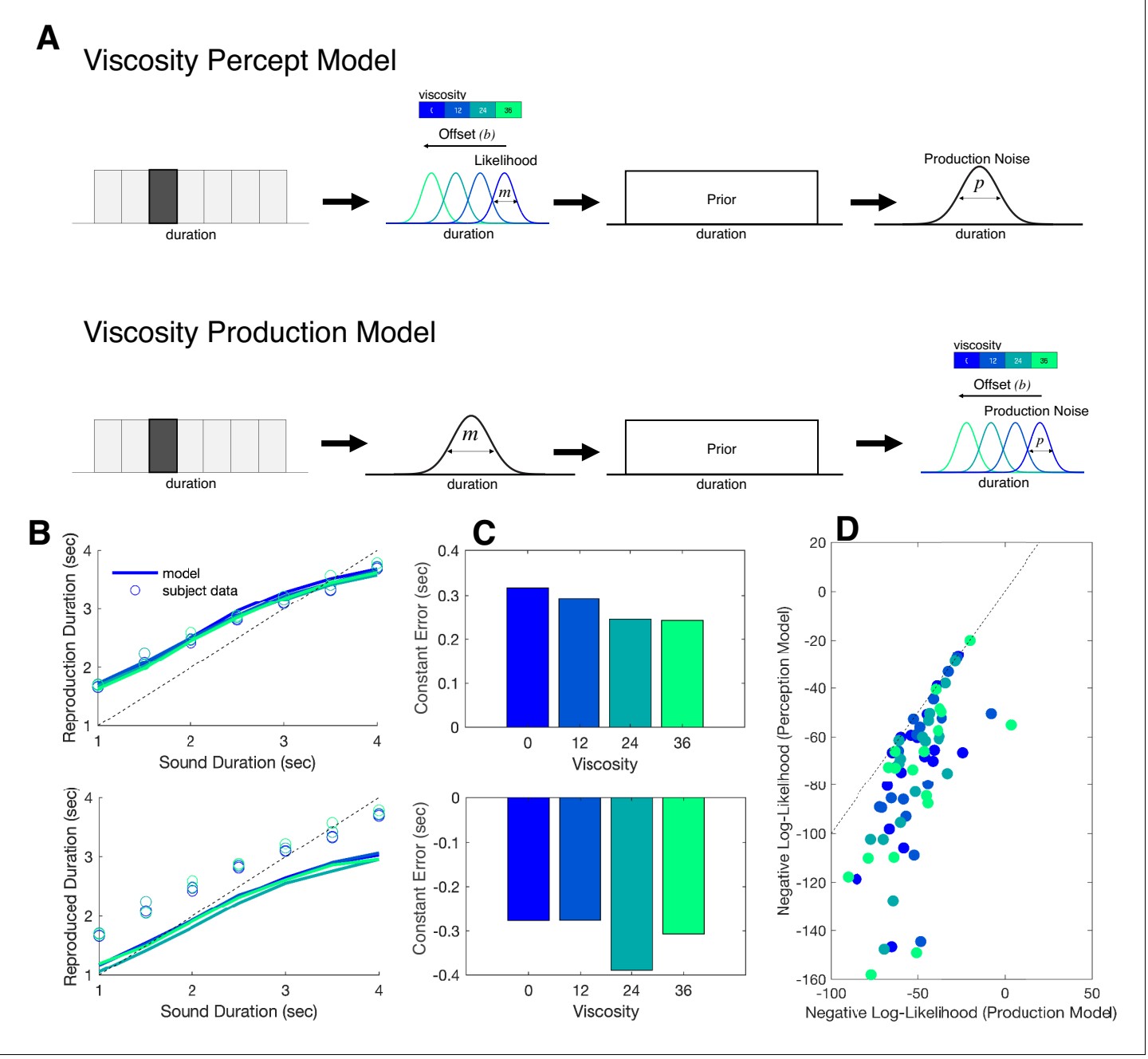

**Figure 7.** Bayesian observer model of reproduction performance and viscosity. (**A**) Schematics for separate Bayesian models, in which the offset term could occur at either measurement level (Perception Model) or the production level (Production Model). (**B**) Predictive checks for both models, in which model simulations are presented with average subject data for Perception Model (top) and Production Model (bottom). (**C**) Average Constant Error for Perception (top) and Production (bottom) model simulations. (**D**) Comparison of Negative Log-Likelihood values for model fits for each model; individual data points represent single subjects, with each color representing a different viscosity. Dashed line represents the identity.

perceptual effect of viscosity on perceived time, we further investigated this effect by administering a temporal reproduction task in Experiment 2. Eliminating the decision process ensured that recorded responses were more representative of timing distortions via perceptual modulation. Here, participants made temporal estimations during movement, and reproduced these via button press. Although in the temporal categorization task the degree of movement between participants was self-selected and highly variable, we attempted to reduce this variability by requiring participants to move continuously during trials in the temporal reproduction task. We also controlled for

performance variability due to familiarization by including a brief training session. Participants exhibited an overestimation bias when reproducing the interval duration, a result previously shown in motor reproduction of auditory intervals (*Shi et al., 2013*). However, the degree of overestimation decreased as a function of viscosity, confirming the compression effect seen in the temporal categorization task. Using a Bayesian observer model (*Jazayeri and Shadlen, 2010*; *Remington et al., 2018*), we observed a linear shift in perceptual bias as a function of viscosity, further supporting a purely perceptual effect of movement slowing on timing.

The results between experiments converge on the finding that viscosity manipulation leads to interval underestimation (reflected in the bias parameters). Further, timing precision was generally unaffected by our manipulations. In the temporal categorization task, the CV remained constant across viscosities; the only notable CV effect was revealed by the median split analysis, in which trials with greater movement led to greater precision for the zero viscosity condition. This was in agreement with our previous study demonstrating a movement-related enhancement in temporal precision estimates (*Wiener et al., 2019*), and suggests that free movement can improve timing precision only in unrestricted movement environments. The temporal reproduction task also showed that viscosity did not affect the variability of estimation (CV), but interestingly, may have been associated with greater uncertainty as indicated by increased central tendency of reproduction slopes.

In addition to the paradigm difference in Experiments 1 and 2, it is worthwhile to consider the methodological differences that may have influenced these results. As mentioned above, in our first experiment we allowed movement, while in the second experiment we required it. Additionally, movement in the temporal categorization task had utilitarian value; participants could strategically approach candidate targets, and thus movement offered the potential to improve performance by shortening RTs. Movement in the temporal reproduction task did not provide this decision-making advantage, but notably the perceptual biases due to viscosity maintained the same directionality across experiments. That is, in response to viscosity, the BP parameter in Experiment one and the perceptual bias parameter in Experiment 2 shifted upwards and downwards respectively, in accordance with temporal compression. This suggests that movement influenced similar temporal estimation mechanisms, and despite methodological differences, the biasing effect of viscous environments was robust under different task demands. One additional note regarding this work is that subjects were not required to time their movements but were rather using their movements for timing. We believe the distinction here is important, as the ancillary movement patterns nevertheless influenced the perceived timing.

Our results parallel prior work investigating temporal distortions as a function of movement parameters. (*Press et al., 2014*) presented tactile stimulation to participants' fingers during movement or while stationary as they viewed congruent or incongruent hand avatars displayed on a screen. The duration of the tactile stimulus was perceived as longer when the avatar was congruent to concurrent movement. (*Yon et al., 2017*) similarly found that when participants executed finger movements of a pre-specified duration and listened to an auditory stimulus with an independently selected duration, judgments were biased towards the duration of the movement. They followed this with a separate experiment focused on manipulating movement distance as a proxy for movement duration (with the knowledge that farther reaches typically take longer); participants reached towards targets along a flat workspace with variable distances while a concurrent auditory duration was presented. Durations presented during 'far' reaches were again perceived as longer. This bears similarity to our experiments here, with a few key distinctions: in our experiments, movements were scaled exclusively in the spatial, but not temporal domain as a function of viscosity (i.e. movement distance, but not movement duration, scaled with viscosity). Additionally, the Yon experiments relied on volitional modulation for task success (e.g. being trained to make 'short' and 'long' movements or reach 'near' and 'far'). Our manipulation affected movement distance without any change in task demands; that is, the manipulated sensory feedback (via the somatosensory system) offered no explicit benefit or detriment to completing the task. Participants certainly reacted to these perturbations by increasing applied force, but self-modulation and monitoring in response to the manipulation was not required as in *Yon et al., 2017*. We believe this is a critical insight given by our study; even though participants knew that demands did not change with viscosity, the manipulation still induced temporal biases – presumably without conscious awareness. Taken together, these findings suggest complementary mechanisms of temporal biasing by movement distance and duration that are often conflated due to distance-duration correlations.

In contrast, some prior accounts do not align with our results. For example, *Yokosaka et al., 2015* found that visual intervals demarcated by pairs of visual flashes were compressed during fast hand movement, whereas in our experiment we show that slowing down movement leads to compression. Additionally, Tomassini and colleagues (*Tomassini et al., 2014*) reported that tactile intervals were compressed during hand movement. Considering these examples, a crucial note–as highlighted in *Iwasaki et al., 2017*–is that many of the distortive effects of movement can be linked to whether an interval is filled or unfilled. Indeed, while we utilized filled auditory intervals in our tasks, the studies with contrasting effects utilized unfilled intervals. Also of interest is the type of movement in the listed studies and the interval ranges used. Movements were typically stereotyped across trials, and intervals were tested in the subsecond range. Here, we allowed participants to move freely along a two-dimensional plane and there were considerable individual differences in selected trajectories. Most notably, it was the *externally* imposed restriction of movement that turned out to be more influential on temporal perception than self-modulated movement characteristics.

In future movement-timing experiments it may be fitting to look beyond simple movement parameters (e.g. speed and distance) and focus on higher-order parameters such as biological versus non-biological motion. Specialized detection systems in the brain can identify movement from other organisms that adhere to physical principles such as natural acceleration and deceleration patterns, and are studied in laboratories from using simple stimuli (e.g. moving dot displays; *Gavazzi et al., 2013*) to complex light-point representations of locomotion (*Wang and Jiang, 2012*). When timing the duration of a dot moving across a screen with different movement profiles, biological motion (compared to constant motion) is timed more precisely for sub- and supra-second intervals, and more accurately for sub-second intervals (*Gavazzi et al., 2013*). *Wang and Jiang, 2012* found that the perceived duration of a human light-point display was expanded compared to a static or non-biological motion display, and this effect persisted when the dot positions were scrambled but retained local kinematics. However, studying motor production of biological motion during timing is a fairly new concept. *Carlini and French, 2014* asked participants to time a dot stimulus moving across a screen that they either tracked with their finger, or viewed passively. Hand tracking overall improved accuracy and precision, but the improvement occurred irrespective of the movement type (biological motion, constant velocity, or sharp 'triangular' velocity profile). Additionally, biological motion was timed most accurately and precisely with or without manual tracking. This highlights a benefit of concurrent movement that can increase the accuracy of non-biological motion timing to biological motion levels. A natural question that follows concerns the degree to which these different motion types are biased from external factors. For example, would the compressing effect of viscosity occur more strongly for non-biological motion because it is more susceptible to motor influences? If we were to administer a similar manual tracking task, we predict that this would be the case. This design would also allow us to examine the separate contributions of force and movement distance; to examine movement distance we would scale the cursor speed down with viscosity (thereby requiring the same amount of cumulative force between viscosities), and to examine force we would keep the speed constant (requiring a compensatory force increase with viscosity).

This novel contribution to the existing body of research highlights the importance of sensory feedback in timing, whereas the study of movement-induced time distortions has focused primarily on feedforward effects. These complementary accounts enrich the current understanding of the coupling of movement and time perception, and support the longstanding notion that interval timing in the brain utilizes multiple streams of sensory information and distributed timing circuits to form a unified percept of duration (*Chen and Vroomen, 2013*; *Bausenhart et al., 2014*; *Wiener et al., 2011*). Understanding the integration of these signals is an important problem in modern neuroscience, and here we have presented a strong case for greater investigation into the role of movement perturbations in time perception. More specifically, they point to an intrinsic role of the motor system in time perception. As described above, previous research has shown motor system involvement in time perception, even when no timed motor response is required (*Nani et al., 2019*). Further, temporal categorization, as employed here has been linked to motor system processing and neural populations within the supplementary motor area (SMA), a region highly implicated in timing (*Schwartze et al., 2012*; *Mendoza et al., 2018*; *Méndez et al., 2014*).

Most saliently, our results align with A Theory of Magnitude (ATOM), an account outlining common neural circuitry between spatial, temporal, and numerosity representations in the brain

(*Walsh, 2003*). The manipulation we introduced scaled spatial characteristics of movement, and subsequently led to a scaling down of perceived duration. A relevant consideration here is that viscosity was introduced via sensorimotor channels but had a cross-modal effect on auditory timing. This highlights an issue not originally included in ATOM: the role of simultaneous temporal measurements from different sensory channels. Our results suggest that the two channels of temporal processing were not redundant; if this had been the case, there would have been no effect of movement on the separate auditory estimate. We propose that these effects can be approached from a perspective of optimal cue combination (*Ball et al., 2017*). In a previous study (*Wiener et al., 2019*), we found that movement (versus no movement) enhanced temporal precision. Combined with the knowledge that movement can also bias timing, we suggest that noisy auditory estimates in our experiments reaped a benefit from integrating motor information with higher precision, at the cost of a slight bias. To more thoroughly understand how motor and sensory time estimates interact, it is essential to examine the weighting of separate sensory pathways according to their usefulness and their reliability. Although the current study is not equipped to dissociate motor and auditory contributions, we strongly recommend this principle as a productive area of focus for future work.

Beyond understanding basic mechanisms of temporal processing and movement, these approaches may be of interest in order to study clinical disorders for which timing and movement are disrupted. Notable work in recent years has strongly suggested that motor control is an extension of ongoing cognitive computations (*Lepora and Pezzulo, 2015*; *Resulaj et al., 2009*), and adopting an integrated view of cognition-action pathways is a promising avenue for understanding these disorders and developing therapies that exploit these links. For example, in the case of Parkinson's (PD) and Huntington's Diseases (HD), core timing networks in the brain overlap substantially with the motor circuitry targeted by neural degeneration, such as the basal ganglia (*Obeso et al., 2000*; *Obeso et al., 2014*; *Browne et al., 1997*). Motor deficits in these diseases are often accompanied by timing deficits (*Avanzino et al., 2016*) and other cognitive abnormalities (*Robbins and Cools, 2014*; *Paulsen, 2011*). The shared neural circuitry combined with these parallel deficits provide a basis for incorporating movement into cognitive training and vice versa.

In contrast, psychiatric and neurodevelopmental disorders are usually discussed in terms of cognitive deficits despite exhibiting motor idiosyncrasies. Although embodied cognition has gained traction in basic science research, there are fewer approaches in clinical research that consider cognitive and motor symptoms in relation to one another. One example is attention deficit hyperactivity disorder (ADHD). The excessive motoric activity (i.e. hyperactivity) associated with the disorder is typically seen as disruptive, but interestingly, some studies have shown that it can boost cognitive control performance (*Rapport et al., 2009*; *Hartanto et al., 2016*). In light of the timing deficits present in ADHD (*Plummer and Humphrey, 2009*), it would be interesting to explore the extent movement can provide a similar benefit to timing performance. The coupling of timing and motor functions taken together with the supramodal nature of core timing circuits provides an excellent opportunity to probe cognition-action pathways in various clinical disorders. On the one hand, movement can sharpen certain perceptual and cognitive processes, and on the other hand, it can introduce biasing effects on timing and on other perceptual judgments (*Moher and Song, 2014*), including when movement perturbations are applied (*Hagura et al., 2017*). Therefore, further investigation is warranted to better understand how these effects can be exploited to improve outcomes for patients.

In summary, we tested participants' ability to time intervals while moving, and in two separate timing paradigms demonstrated that imposing restrictions on movement subsequently shortened perceived time. Computational modeling confirmed that these effects arose from perceptual differences rather than downstream cognitive processes. The influence of motor activity on sensory processing is well studied–motor activity can modulate sensory processing across modalities, as early as in primary sensory cortices (for review, see *Parker et al., 2020*). For example, locomotion can modulate gain in rodent visual and auditory cortices (*Niell and Stryker, 2010*; *Zhou et al., 2014*), and in humans, orientation change detection is improved when preparing to make a grasping motion aligned with the original orientation (*Gutteling et al., 2011*). Recent reports have shown that timing, through a neurally distributed system, is also modulated by and can even be improved by movement (*Wiener et al., 2019*; *Carlini and French, 2014*; *Manning and Schutz, 2013*). Our work confirms that time is subject to distortion by externally imposed movement constraints and allowances, at least in the auditory modality and within the suprasecond range. It is important to note that timing distortions can be modality-specific (*Bueti, 2011*), and as stated above, differ when intervals are

filled versus unfilled (*Iwasaki et al., 2017*). These are natural considerations regarding our results, and could form the basis for future work to investigate different modalities, temporal ranges, and interval presentation styles. Another consideration is that in our experiments, viscosity was fixed throughout each trial; in followup experiments, it would be interesting to calibrate the nature and degree of movement perturbations to observe the resulting effects on timing. For example, experimenters could introduce dynamic perturbations or alter visual feedback much like in motor adaptation experiments (*Shadmehr and Mussa-Ivaldi, 1994*; *Krakauer et al., 2000*; *Alhussein et al., 2019*; *McKenna et al., 2017*; *Zhou et al., 2017*).

## Materials and methods

### Participants

A total of 28 participants took part in Experiment 1 (18 female, 10 male, M age = 23.5 (7.0)) and 18 separate participants took part in Experiment 2 (7 female, 11 male, M age = 21.5 (4.1)) for $15 per hour in gift card credit. These sample sizes were chosen to accord with our previous report (*Wiener et al., 2019*). All participants were right-handed as measured by the Edinburgh Handedness Inventory (*Oldfield, 1971*). All protocols were approved by the Institutional Review Board at the University of California, Davis.

### Apparatus

Both experiments utilized a robotic arm manipulandum (KINARM End-Point Lab, BKIN Technologies; *Nguyen et al., 2019*; *Hosseini et al., 2017*). Here, the participants manipulated the right arm on a planar workspace to perform the tasks and were blocked from viewing their arm directly by a horizontal screen display. A downward-facing LCD monitor reflected by an upward-facing mirror allowed viewing of experiment start locations and targets, demarcated by small circles. Participants were seated in an adjustable chair so that they could comfortably view the mirrored display. In Experiment 1, a cursor on the screen projected their current arm position during each trial, whereas in Experiment 2 no cursor was present. The manipulandum sampled motor output at 1000 Hz.

### Procedure

#### Experiment 1

In the first experiment, participants performed an auditory temporal categorization procedure (*Kopec and Brody, 2010*) with intervals of 1000, 1260, 1580, 2000, 2520, 3170, and 4000 ms with a 440 Hz tone. A total of 280 trials were segmented into five blocks with the option for a short (1–2 min) break between each block. Participants were instructed to start each trial in a central target location, where the manipulandum locked the arm in place for 1000 ms. A warm-up phase began as the hold was released and the words 'Get Ready' were displayed on the screen. Participants were encouraged during instruction to move freely throughout the workspace during each trial, and respond as quickly and accurately as possible. During the warm-up phase, viscosity was applied in a linearly ramping fashion, reaching one of four viscosity values (0, 12, 24, or 36 Ns/m$^2$) in 2000 ms. Simultaneously, two response targets appeared at 105° and 75°, equidistant from the starting location. Target assignment was counterbalanced between participants. Once the desired viscosity was reached, the tone began to play and participants were required to determine whether the tone was short or long compared to all tones they had heard so far (reference-free categorization) by moving the cursor to the corresponding target location on the right or left. If a response was made before the tone had elapsed, the trial was discarded and they were required to repeat the trial. No feedback was given. Viscosity and duration values were randomized across trials with equal representation in each block.

#### Experiment 2 (temporal reproduction)

In the second experiment, a separate group of participants performed a temporal reproduction task with tone intervals of 1000, 1500, 2000, 2500, 3000, 3500, and 4000 ms. Because the task had higher attentional demands and was more likely to cause fatigue, the 280 trials were segmented into 10 instead of five blocks. In this task, the manipulandum moved the participant's arm (1000 ms) to the one of 16 encoding start locations in a grid-like configuration, and locked in place for 1000 ms until

a green 'go' cue appeared in the start location. Upon seeing the cue, participants were required to start moving. Moreover, the tone onset was contingent upon movement, and the trial was discarded and repeated if movement stopped before tone offset. After tone offset, a linearly ramping 'brake' was applied to discourage further movement. Once movement stopped, the manipulandum moved to a central location for the reproduction phase. After seeing a green cue, participants reproduced the encoded interval by holding and releasing a button attached to the manipulandum. No auditory or performance feedback was given. As in Experiment 1, duration and viscosity were randomized and equally represented in each block.

## Analysis

In Experiments 1 and 2, movement distance and force measures were taken for each trial. Movement distance was defined as the summed distance traveled (point-by-point Euclidean distance between each millisecond time frame) during the stimulus tone. Force was similarly defined as the summed instantaneous force during the stimulus tone. In Experiment 1, RT was defined as the time elapsed between tone offset and reaching one of the two choice targets. Outlier trials were excluded for RT values greater than three standard deviations away from the mean of a participant's log-transformed RT distribution (*Ratcliff, 1993*). For each participant we plotted duration by proportion of 'long' responses. From here, we used the psignifit 4.0 software package to estimate individual BP and coefficients of variation (CV) for all four viscosity values (*Schütt et al., 2016*); all curves were fit with a cumulative Gumbel distribution to account for the log-spaced nature of tested intervals (*Wiener et al., 2018*; *Wiener et al., 2019*). The BP was defined as the 0.5 probability point on the psychometric function for categorizing intervals as 'long'; the CV was defined as half the difference between 0.75 and 0.25 points on the function divided by the BP.

In Experiment 2, we plotted true duration by estimated duration for each participant to find individual slope and intercept values. We also computed individual CV values for duration and viscosity conditions via the ratio of estimation standard deviation to estimated mean (*Wiener et al., 2019*). We excluded trials if the reproduction time fell outside three standard deviations from the mean. Additionally, we calculated the constant error, defined as the difference between the reproduced interval and the presented one.

For statistical analyses, we report the results of repeated measures ANOVAs and post-hoc t-tests where appropriate, along with effect sizes. For all correlations, we report the value of both Pearson and Spearman correlation coefficients. Additionally, for effects deemed non-significant by standard null-hypothesis statistical tests, we report Bayes Factors as calculated by the software package JASP (http://www.jasp-stats.org).

## Drift diffusion modeling

To better dissect the results of Experiment 1, we decomposed choice and RT data using a drift diffusion model (DDM; *Ratcliff, 1978*; *Wiecki et al., 2013*; *Tipples, 2015*). Due to the low number of trials available per condition, we opted to use hierarchical DDM (HDDM) as employed by the HDDM package for Python (http://ski.clps.brown.edu/hddm_docs/allinone.html). In this package, individual subjects are pooled into a single aggregate, which is used to derive fitted parameters by repetitive sampling from a hypothetical posterior distribution via Markov Chain Monte Carlo (MCMC) sampling. From here, the mean overall parameters are used to constrain estimates of individual-subject estimates. HDDM has been demonstrated as effective as recovering parameters from experiments with a low number of trials (*Wiecki et al., 2013*).

In our approach to modeling, we opted for a DDM in which four parameters were set to vary: the threshold difference for evidence accumulation ($a$), the drift rate towards each boundary ($v$), the starting point, or bias toward a particular boundary ($z$), and the non-decision time ($t$), accounting for remaining variance due to non-specific processes (e.g. perceptual, motor latencies). Our choice to include these four parameters was driven a priori by earlier modeling efforts for studies of timing and time perception (*Tipples, 2015*; *Balcı and Simen, 2014*; *Wiener et al., 2018*). Nevertheless, we compared fits of models of varying complexity via the Deviance Information Criterion, in which parameters added in the following manner: null -> v -> va -> vat -> vatz. A reduction in DIC of 10 or greater was considered as evidence for a better fit. We additionally include between-trial variability in the starting point (sz) to account for trial-to-trial fluctuations.

For model construction, we relied on the model of *Balcı and Simen, 2014* as a guide, as in our previous work (*Wiener et al., 2018*; *Bader et al., 2019*). In this model, performance on a temporal categorization task is conceived as a two-stage process. In the first stage, evidence accumulates monotonically during the interval, serving as a measure of elapsed time. At interval offset, the second stage initiates another drift-diffusion process, in which evidence accumulates towards one of two decision bounds, associated with 'short' and 'long' categorizations. Notably, the starting point (z) of the second stage process is determined by the level of accumulation from the first stage at offset; shorter intervals lead to a starting point closer to the 'short' categorization boundary, whereas longer intervals are closer to the 'long' boundary. Further, the proximity to each of the boundaries also determines the drift-rate (z) of the second stage process, with higher drift rates for closer proximity. Thus, the drift rate and the starting point are linked in this model.

Following our previous work, we chose to model only the second-stage process. This choice comes from previous studies as well as Balcı and Simen (2014; supplementary material) who demonstrated that modeling the second-stage alone is sufficient at capturing all of the predicted effects. However, in designing the model, we note that there was an insufficient number of trials to account for both duration and viscosity, as the number of trials for each combination was lower than recommended by HDDM ($\leq$10). As such, in order to demonstrate that this model could account for the behavioral data, we built two DDMs – one in which the included parameters varied as a function of duration, and one in which they varied as a function of viscosity.

Model construction was accomplished using the HDDMStimCoding class for HDDM, in which the starting point was split between both short and long response boundaries. Unlike the behavioral analysis, we included all trials here, and chose to model the probability of outliers using the *p_outlier* option, in which outliers were assumed to come from a uniform distribution at the right tail of the full RT distribution (*Ratcliff and Tuerlinckx, 2002*). This was done to avoid differential weighting of RTs from individual subjects in the full distribution; the probability was set to 0.05. Model sampling was conducted using 10,000 MCMC samples, with a burn-in of 1000 samples and a thinning (retention) of every 5th sample. Individual model fits were assessed by visual inspection of the chains and the MC_err statistic; all chains exhibited low autocorrelation levels and symmetrical traces. We additionally sampled five further chains of 5000 iterations (200 burn-in) of the 'winning' models and compared the Gelman-Rubin Statistic (*Gelman and Rubin, 1992*) revealing a value of 1.011 ± 0.082 (SD) for the Viscosity model, and 1.04 ± 0.093 (SD) for the Duration model, indicating good chain stability. In addition, we conducted posterior predictive checks for both models, in which 500 samples from the posterior distributions of each parameter were randomly drawn and used to generate a new dataset. The resulting choice and RT data were analyzed and compared to behavior for both duration and viscosity models.

Once model fits were accomplished and compared to behavioral data, we sought to demonstrate that a 'full' model – one in which duration and viscosity varied – could account for the behavioral findings. To accomplish this, we combined both models by first conducting a posterior predictive check for the Duration model, and then shifting each drawn sample from the posterior by an amount determined by the Viscosity model for each of the four levels of viscosity. For example, the drift rate (v) drawn from the posterior of the duration model would be averaged with the drift rate from the Viscosity model; a new dataset would then be generated using these values. In this way, four separate datasets from the Duration model were simulated, one for each viscosity. Choice and RT data were then averaged in these datasets to see if they recapitulated the original findings.

Finally, we also performed a non-hierarchical fit to subject data. This was done to confirm that the hierarchical results and their correlation with behavior were not unduly influenced by shrinkage of parameter estimates to the group mean. To accomplish this, individual subject data were fit using the HDDM.Optimize() function, in which results were fit via a Maximum Likelihood procedure (*Ratcliff and Tuerlinckx, 2002*). Individual parameter estimates were compared to the hierarchical ones, and the same correlations with behavior were conducted. This comparison was only carried out for the Viscosity model, where correlations with behavior were conducted.

## Bayesian observer model

To model data from the reproduction task, we employed a Bayesian Observer Model, as developed by Jazayeri and colleagues (*Jazayeri and Shadlen, 2010*; *Remington et al., 2018*). In this model, sensory experiences of duration are treated as noisy estimates from a Gaussian distribution with

scalar variability that grows linearly with the base interval, termed the measurement noise ($m$). Once drawn, these estimates are combined with the prior distribution of previously-experienced intervals; in this case, the prior was modeled as a uniform distribution with an upper and lower boundary corresponding to the presented intervals in the task. The mean of the resulting posterior distribution of an interval is thus drawn to the mean of the prior, thus accounting for the central tendency effect observed. Further, this effect also accounts for a trade-off in the precision of estimates; increased reliance on the prior, while increasing bias to the mean, also reduces variability, thus decreasing the CV (*Cicchini et al., 2012*). Following the posterior estimate, the produced movement is additionally corrupted by movement noise ($p$), again drawn from a Gaussian distribution. As an additional parameter, measurement bias is also included ($b$), also termed the estimation 'offset' (*Remington et al., 2018*), in which the noisy estimate is shifted away from the true duration. Note here that $b$ is specifically included as a shift in perception, rather than production bias, and so we refer to this as the Perception Model.

We additionally constructed a second, alternative version of this model, in which the offset parameter was instead shifted to the production stage. Specifically, the offset term was changed in this model to be added during production, following movement noise, referred to as the Production Model. Model parameters ($m$, $p$, $b$) for each model were fit by minimizing the negative log-likelihood of individual subjects' single trial responses, using modified code provided at (https://jazlab.org/resources/). Minimization was accomplished using the *fminsearch* function for Matlab, using numerical integration over the posterior distribution. Model fits were repeated using different initialization values and a fitting maximum of 3000 iterations; inspection of fitted parameters indicated good convergence of results. Model comparison was conducted by comparing negative log-likelihood values across each of the four viscosity conditions.

Lastly, we conducted predictive checks by taking the average parameter estimates across subjects and simulating two datasets (40 trials per condition). These datasets were then analyzed in a similar manner to the behavioral data and compared to average subject data. Separate simulations were conducted for the Perception and Production Models.

## Acknowledgements

This work was supported by the National Science Foundation (1849067), awarded to MW and WMJ.

## Additional information

### Funding

| Funder | Grant reference number | Author |
|---|---|---|
| National Science Foundation | 1849067 | Martin Wiener |

The funders had no role in study design, data collection and interpretation, or the decision to submit the work for publication.

### Author contributions

Rose De Kock, Conceptualization, Data curation, Formal analysis, Investigation, Visualization, Writing - original draft; Weiwei Zhou, Data curation, Formal analysis, Investigation, Methodology; Wilsaan M Joiner, Resources, Software, Supervision, Funding acquisition, Methodology, Project administration, Writing - review and editing; Martin Wiener, Conceptualization, Data curation, Formal analysis, Supervision, Funding acquisition, Validation, Investigation, Visualization, Methodology, Writing - original draft, Project administration, Writing - review and editing

### Author ORCIDs

Martin Wiener ![ORCID] https://orcid.org/0000-0001-5963-5439

### Ethics

Human subjects: Informed consent was obtained from all subjects. All protocols were approved by the Institutional Review Board at the University of California, Davis (IRB Protocol # 1336438-6).

## Decision letter and Author response

Decision letter https://doi.org/10.7554/eLife.63607.sa1
Author response https://doi.org/10.7554/eLife.63607.sa2

# Additional files

## Supplementary files

• Transparent reporting form

## Data availability

All source data have been deposited in Dryad. Located at https://doi.org/10.25338/B8S913.

The following dataset was generated:

| Author(s) | Year | Dataset title | Dataset URL | Database and Identifier |
|---|---|---|---|---|
| Kock R, Zhou W, Joiner WM, Wiener M | 2021 | Slowing the Body slows down Time (Perception) | https://doi.org/10.25338/B8S913 | Dryad Digital Repository, 10.25338/B8S913 |

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
