## [Decision Letter]

**Acceptance summary:**

This study investigates the effects of viscous movement environments on temporal processing. The results show that systematically impeding movement during interval timing leads to a subsequent compression of perceived duration in multiple tasks. Computational modeling shows that the effect of movement is directly on time perception and not on decision-making. This is an interesting and well performed study, providing novel insights into the interaction between time perception and reaching movements affected by viscous environments.

**Decision letter after peer review:**

Thank you for submitting your article "Slowing the Body slows down Time (Perception)" for consideration by *eLife*. Your article has been reviewed by 3 peer reviewers, including Hugo Merchant as the Reviewing Editor and Reviewer #1, and the evaluation has been overseen by Michael Frank as the Senior Editor. The following individual involved in review of your submission has agreed to reveal their identity: Clare Press (Reviewer #3).

The reviewers have discussed the reviews with one another and the Reviewing Editor has drafted this decision to help you prepare a revised submission.

As the editors have judged that your manuscript is of interest, but as described below extensive revisions are required before it is published, we would like to draw your attention to changes in our revision policy that we have made in response to COVID-19 (https://elifesciences.org/articles/57162). First, because many researchers have temporarily lost access to the labs, we will give authors as much time as they need to submit revised manuscripts. We are also offering, if you choose, to post the manuscript to bioRxiv (if it is not already there) along with this decision letter and a formal designation that the manuscript is "in revision at *eLife*". Please let us know if you would like to pursue this option. (If your work is more suitable for medRxiv, you will need to post the preprint yourself, as the mechanisms for us to do so are still in development.)

Summary:

This study investigates the effects of viscous movement environments on temporal processing during a single interval categorization or a reproduction task using durations in the range of seconds. Healthy volunteers timed auditory intervals while moving a robotic arm that randomly applied four levels of viscosity. The results showed a consistent decrease in the bisection point (or point of subjective equality) in the categorization task and a decrease in duration of the produced intervals in the reproduction task as a function of viscosity. A drift-diffusion model and a Bayesian observer model suggested that the underestimation of intervals in the two tasks are due to changes in time perception instead of biases in decision making.

Essential revisions:

This is an interesting and well performed study, providing novel notion on the interaction between time perception and reaching movements affected by viscous environments. Nevertheless, we have a set of serious concerns that need to be addressed in the present version of the paper, as follows:

1. As the authors outline, previous studies have shown that a variety of movement features impact upon duration judgements. The novelty of their study is suggested to be that movement is restricted by environmental factors rather than 'self' modulations. E.g., the finding reflected in the title of "slowing the body slows down time" has already been demonstrated, by my understanding, in four experiments by Yon et al. (2017, JEPG). However, by my reading both in previous studies and the present studies in fact it is environmental modulations that result in self modulations. I.e., I did not comprehend the difference and this needs clarifying for the theoretical value of these studies to be understood. E.g., previous studies have generated the cells in their design by presenting targets at different locations (e.g., Yon et al., 2017 – external modulation) and participants then choose to move to those targets (volitional component). The present studies create the cells by imposing different forces upon the lever while keeping the targets in the same location. In all studies they are choosing to move to targets at different places and under specific environmental viscosities. The authors describe using sensory feedback due to these environmental features but they would use sensory feedback to monitor both visual target location and forcefields – both of which are constant during a specific movement in previous and present work. Once this line is clarified, what would be the assumed different mechanisms involved in the two cases?

2. The values on bisection point show enormous individual differences between subjects, with a range between 1.5 to 3 seconds, why? In contrast, the effect of viscosity on the BP is very subtle and a large proportion of subject showed negative slope of BP as a function of viscosity (please report). This means that in a subgroup of subjects, an increase in viscosity was accompanied with an increase instead of a decrease in the length of perceived time. This finding does not support one of the mayor conclusions of the paper.

3. In the present studies a number of features of the movement are confounded as viscosity is manipulated. As viscosity increases, the length traversed goes down in a given period of time, the force goes up, the speed goes down. As far as I could see the authors do not disentangle which movement feature was responsible for the influence on duration judgements. My guess is that the correlations are high between these features but perhaps there is enough natural variation in the dataset to disentangle. Could a regression analysis be performed to establish the unique contributions of the different movement parameters? As well as helping the reader to understand what is driving the effects it may resolve some of the inconsistencies with respect to previous literature. E.g., the authors observe that fast movements are associated with increased duration judgements, in contrast with some previous findings. They explain the difference as possibly driven by the fact the conflicting studies have unfilled intervals. However, they do not explain why this may alter the operation of the mechanism. Additionally, there is an experiment in Yon et al. (2017) where high speed is associated with compressed estimates and the intervals are filled here.

4. Based on DDM fits authors claim that the effect of viscosity was perceptual (and not through decision-biases). This claim is based on the observation that the viscosity level did not affect the starting point or threshold setting but it exclusively modulated the drift rate. Although on the face value, this appears as a valid conclusion based on the model fit results, the methodological details actually put this conclusion into question by invalidating the model fits (unless participants could predict the next viscosity level – see alternative explanations below). Let me elaborate more on this. The standard assumption in the conventional DDM that authors also used in their analysis is that the threshold is set adaptively in response to the context the participant is in (e.g., taking into account the response-to-stimulus interval, reward/penalty ratio, whether instructions emphasize speed or accuracy, prior probability of different alternatives). In other words, the decision threshold and starting point of the decision process are modulated in response to some average representation of the task statistics or in response to a discriminative stimulus that allows participants to predict the characteristics of the next trial (e.g., a trial with vs. without penalty for errors or trial in which the viscosity will be higher vs. lower). But in the current study, the viscosity levels were manipulated randomly from one trial to the next (without any discriminative signaling) and thus participants could not have adjusted the decision thresholds or starting points (unless authors are willing to step out of the conventional DDM domain in which case they have to assume within-trial dynamic modulation of these parameters, which is rather debatable – see Karsilar et al. (2014). Speed Accuracy Trade-off Under Response Deadlines. Frontiers in Neuroscience, 8:248. doi3: 10.3389/fnins.2014.00248). Thus, for DDM to inform us regarding the source of the observed effects, one needs a blocked design where the viscosity levels are kept constant within a given block. What the current results tell me is the following; Yes, the effect of movement on temporal judgments can be mediated by the perceptual effects. What they do not tell me is the following; Movement does not affect the other components of the decision process (for which authors need a different experimental design). I recommend authors to look at Bogacz et al. (2006). The physics of optimal decision making: a formal analysis of models of performance in two-alternative forced-choice tasks. Psychological Review, 113(4), 700-765. https://doi.org/10.1037/0033-295X.113.4.700 for relevant discussions (look at the sections on the optimal performance).

But there is one way out based on the sequential nature of the diffusion process in the temporal bisection task. At least as it is formulated in Balcı and Simen (2014), which authors cite in the paper, the decision process that would be captured by conventional DDM fits is the second stage diffusion process the parameters of which are contingent on the first stage diffusion (opponent Poisson DDM) process. Within this framework, authors can argue that the viscosity experienced during the movement during the timing of the test stimulus (prior to the offset of the stimulus) can actually modulate the relevant parameters of the second stage diffusion before its launching. If authors are willing to take this route, then they should be mechanistically savvier about such contingencies; otherwise, the results and assumptions are not tied to a concrete process and thus lack the necessary theoretic grounding (as they currently are in the paper). I recommend authors to let the model tie things together, which will bolster the theoretical cohesion of the work.

I have concerns regarding the use of latent parameter estimates from the HDDM (or slopes estimates gathered from their regression on a factor) in the correlation analysis. As the authors state in the Methods section, HDDM uses the mean overall parameters to constrain the estimates for individual subjects; this is a great analytical tool to increase the statistical power for between-group comparisons but at the same time it makes the estimates for individual subjects unsuitable for correlation analysis. If possible given the number of data points, authors should fit the individual data for estimates to be used in the correlation analyses.

I have another concern regarding the theoretical handling of the application of DDM to the data. Using DDM in the temporal bisection task actually requires assuming a sequential diffusion process (see Comment 1). This has been previously formulated in Balci and Simen, (2014). Fitting conventional DDM is just an approximate method to capture the behavioral output of the sequential process. This sequential process requires yoking the drift-rate and starting point during model fits or testing the relationship between them to validate the model in the context of the dataset at hand. Based on this model, if one assumes that viscosity of the movement affects the perceptual/timing process (as hypothesized in this work) then one would not only expect a relationship between viscosity and the drift rate but also a relation between viscosity and the starting point. This issue should be addressed in the paper.

5. The quality of DDM fits are not shown in the paper. The quality of the fits should be certainly demonstrated for both models. Please provide an example of the actual and predicted DDM psychometric and chronometric functions.

6. The fact that the model parameters (DDM and Bayesian) show a non-linear relation with viscosity does not imply that they could also have a role in explaining the behavior. A downstream non-linear reader could produce the observed behavioral effects of viscosity. Since the changes in DDM threshold and bias as a function of viscosity are nonlinear, it would be interesting to correlate the individual slope of BP x viscosity with the negative coefficient on an exponential or power function fitted to these parameters vs viscosity.

7. The authors suggest that the DDM results support the notion that the shift in BP produced by larger viscosities is due to a decrease in drift rate, which in turn produced changes in the perceived duration rather than decision parameters. However, the authors also suggest that time perception and decision making are intermingled in the bisection task and that once the categorical boundary has been crossed, subjects stop accumulating temporal information (Wiener and Thompson, 2015). How do you reconcile the two results?

8. Although I appreciate the Bayesian Observer Model (as I indicated at the very beginning of my review), I think this is not empirically well-justified given the lack of an effect on the level of CV. Do authors assume that the uncertainty characteristics of the representation itself and reproduction (motor component) are independent enough (I see that they are not) but even in that case, the problem continues since the motor output would inevitably inherits noise that results from the representational uncertainty. Consequently, I think that the motivation behind this cool analysis, with contemporary relevance to the interval timing literature, should be buttressed with stronger empirical or theoretical justifications (of course, while making sure of not being over-speculative).

9. It would be interesting if the authors could comment upon the precise mechanisms via which each of these features may shape duration perception. Presumably an influence of movement duration on duration perception is easier to explain mechanistically than an influence of distance per se? Why would distance be thought to provide information concerning auditory duration, independently from the distance-duration correlation? E.g., by my understanding size-duration relationships often observed are thought to be mediated by statistical relationships in natural environments, but such that independent information of magnitude would not add to that provided by duration (e.g., Vincent Walsh, TICS, 2003).

10. Authors talk about the self-regulation of movement vs. induced constraints on the movement and at times they allude to the embodied cognition. This brings a relevant issue to my mind; within the semi-theoretical stance taken in the paper what would the predictions be for the effect of observed biological vs. non-biological movement on time perception and how do such predictions compare to the results reported in the literature? This could be one way in which the paper can set a stronger theoretical foundation and consolidate some of the other findings in the literature under an overarching theoretical framework in light of the current findings reported in the paper. I see this as a great opportunity for the authors/paper.

Reviewer #1:

This study investigates the effects of viscous movement environments on temporal processing during a single interval categorization or a reproduction task using durations in the range of seconds. Healthy volunteers timed auditory intervals while moving a robotic arm that randomly applied four levels of viscosity. The results showed a consistent decrease in the bisection point (or point of subjective equality) in the categorization task and a decrease in duration of the produced intervals in the reproduction task as a function of viscosity. A drift-diffusion model and a Bayesian observer model suggested that the underestimation of intervals in the two tasks are due to changes in time perception instead of biases in decision making.

This is an interesting and well performed study, providing novel notion on the interaction between time perception and reaching movements affected by viscous environments. Nevertheless, I have a set of serious concerns that need to be addressed in the present version of the paper, as follows:

1. The values on bisection point show enormous individual differences between subjects, with a range between 1.5 to 3 seconds, why? In contrast, the effect of viscosity on the BP is very subtle and a large proportion of subject showed negative slope of BP as a function of viscosity (please report). This means that in a subgroup of subjects, an increase in viscosity was accompanied with an increase instead of a decrease in the length of perceived time. This finding does not support one of the mayor conclusions of the paper.

2. The fact that the model parameters (DDM and Bayesian) show a non-linear relation with viscosity does not imply that they could also have a role in explaining the behavior. A downstream non-linear reader could produce the observed behavioral effects of viscosity. Since the changes in DDM threshold and bias as a function of viscosity are nonlinear, it would be interesting to correlate the individual slope of BP x viscosity with the negative coefficient on an exponential or power function fitted to these parameters vs viscosity.

3. The authors suggest that the DDM results support the notion that the shift in BP produced by larger viscosities is due to a decrease in drift rate, which in turn produced changes in the perceived duration rather than decision parameters. However, the authors also suggest that time perception and decision making are intermingled in the bisection task and that once the categorical boundary has been crossed, subjects stop accumulating temporal information (Wiener and Thompson, 2015). How do you reconcile the two results?

4. It is peculiar that the effect of viscosity was on the produced rather than the measured noise term of the Bayesian model, since the viscosity changes were only present during the interval encoding. Please discuss.

5. The drift rate can be associated with the likelihood function of the measured time of the Bayesian model, whereas the starting point (z) can be linked to the offset shift (b). However, the results showed that viscosity affects the drift rate in the bisection task, whereas affects the offset in the reproduction task. Please discuss why.

Reviewer #2:

The current paper by De Kock et al. investigates the effect of movement on time perception. The novelty of the work is that it considered the effect of externally modulated movements (through manipulating viscosity) instead of self-modulated movements (as done in earlier research) as well as the use of generative and descriptive models to elucidate the mechanisms that underlie such effects. Participants were tested in the temporal bisection (Experiment 1) and temporal reproduction tasks (Experiment 2 – presumably to eliminate the decision component). The consistent result was that higher viscosity primarily resulted in the compression of perceived time (in terms of the rightward shifts in the PSE in Experiment 1 and weaker over reproduction in Experiment 2). The results of DDM fits in Experiment 1 showed that these effects were due to the modulation of the perceptual rather than the decision components. This conclusion was further supported by the effect of viscosity on the perceptual bias parameter (i.e., reduction) of the Bayesian Observer Model fits in Experiment 2. The experiments conducted in this study are certainly interesting and offer a number of novelties; I think that the paper is potentially impactful. That being said, I also have a number of concerns about the work, which I specify below.

1. Based on DDM fits authors claim that the effect of viscosity was perceptual (and not through decision-biases). This claim is based on the observation that the viscosity level did not affect the starting point or threshold setting but it exclusively modulated the drift rate. Although on the face value, this appears as a valid conclusion based on the model fit results, the methodological details actually put this conclusion into question by invalidating the model fits (unless participants could predict the next viscosity level – see alternative explanations below). Let me elaborate more on this. The standard assumption in the conventional DDM that authors also used in their analysis is that the threshold is set adaptively in response to the context the participant is in (e.g., taking into account the response-to-stimulus interval, reward/penalty ratio, whether instructions emphasize speed or accuracy, prior probability of different alternatives). In other words, the decision threshold and starting point of the decision process are modulated in response to some average representation of the task statistics or in response to a discriminative stimulus that allows participants to predict the characteristics of the next trial (e.g., a trial with vs. without penalty for errors or trial in which the viscosity will be higher vs. lower). But in the current study, the viscosity levels were manipulated randomly from one trial to the next (without any discriminative signaling) and thus participants could not have adjusted the decision thresholds or starting points (unless authors are willing to step out of the conventional DDM domain in which case they have to assume within-trial dynamic modulation of these parameters, which is rather debatable – see Karsilar et al. (2014). Speed Accuracy Trade-off Under Response Deadlines. Frontiers in Neuroscience, 8:248. doi3: 10.3389/fnins.2014.00248). Thus, for DDM to inform us regarding the source of the observed effects, one needs a blocked design where the viscosity levels are kept constant within a given block. What the current results tell me is the following; Yes, the effect of movement on temporal judgments can be mediated by the perceptual effects. What they do not tell me is the following; Movement does not affect the other components of the decision process (for which authors need a different experimental design). I recommend authors to look at Bogacz et al. (2006). The physics of optimal decision making: a formal analysis of models of performance in two-alternative forced-choice tasks. Psychological Review, 113(4), 700-765. https://doi.org/10.1037/0033-295X.113.4.700 for relevant discussions (look at the sections on the optimal performance).

But there is one way out based on the sequential nature of the diffusion process in the temporal bisection task. At least as it is formulated in Balcı and Simen (2014), which authors cite in the paper, the decision process that would be captured by conventional DDM fits is the second stage diffusion process the parameters of which are contingent on the first stage diffusion (opponent Poisson DDM) process. Within this framework, authors can argue that the viscosity experienced during the movement during the timing of the test stimulus (prior to the offset of the stimulus) can actually modulate the relevant parameters of the second stage diffusion before its launching. If authors are willing to take this route, then they should be mechanistically savvier about such contingencies; otherwise, the results and assumptions are not tied to a concrete process and thus lack the necessary theoretic grounding (as they currently are in the paper). I recommend authors to let the model tie things together, which will bolster the theoretical cohesion of the work.

2. I have concerns regarding the use of latent parameter estimates from the HDDM (or slopes estimates gathered from their regression on a factor) in the correlation analysis. As the authors state in the Methods section, HDDM uses the mean overall parameters to constrain the estimates for individual subjects; this is a great analytical tool to increase the statistical power for between-group comparisons but at the same time it makes the estimates for individual subjects unsuitable for correlation analysis. If possible given the number of data points, authors should fit the individual data for estimates to be used in the correlation analyses.

3. I have another concern regarding the theoretical handling of the application of DDM to the data. Using DDM in the temporal bisection task actually requires assuming a sequential diffusion process (see Comment 1). This has been previously formulated in Balci and Simen, (2014). Fitting conventional DDM is just an approximate method to capture the behavioral output of the sequential process. This sequential process requires yoking the drift-rate and starting point during model fits or testing the relationship between them to validate the model in the context of the dataset at hand. Based on this model, if one assumes that viscosity of the movement affects the perceptual/timing process (as hypothesized in this work) then one would not only expect a relationship between viscosity and the drift rate but also a relation between viscosity and the starting point. This issue should be addressed in the paper.

4. The quality of DDM fits are not shown in the paper. The quality of the fits should be certainly demonstrated.

5. Authors talk about the self-regulation of movement vs. induced constraints on the movement and at times they allude to the embodied cognition. This brings a relevant issue to my mind; within the semi-theoretical stance taken in the paper what would the predictions be for the effect of observed biological vs. non-biological movement on time perception and how do such predictions compare to the results reported in the literature? This could be one way in which the paper can set a stronger theoretical foundation and consolidate some of the other findings in the literature under an overarching theoretical framework in light of the current findings reported in the paper. I see this as a great opportunity for the authors/paper.

6. I found that the paper could be a bit more inclusive in terms of the previous work on the relationship between movement and time perception. As it stands, the paper does not fully "exploit" the relevant literature to set the stage for the reader.

7. "Entry into the response location prior to the tone offset was penalized by restarting the trial, and so the optimal strategy was to move the cursor closer to the "short" location, and then gradually move to the "long" location as the tone elapses (Wiener et al., 2019)." This happens to be an emergent action pattern in these tasks; do authors think that differential reinforcement was necessary for the emergence of this strategy? It would also be interesting (but not necessary) to look at the parametrization this action pattern in relation to the optimization of temporal decision making under uncertainty.

8. There are several tests in the paper, where the null effects are theoretically important. Those frequentist tests should be coupled with Bayesian counterparts.

9. Figure 1b (left side): One thing I don't understand here is how the hand can reach to the same location for all test durations prior to the termination of test duration. Am I misinterpreting the figure? E.g., Maybe different sigmoidal curves do not correspond to different test durations.

10. Figure 4b bottom panel should be better explained (e.g., explain what slope refers to in the figure caption).

11. Authors should better justify the inclusion of certain parameters in the model (e.g., T_naught_). On a related note, the proper model comparisons should be exercised; for instance, models with different levels of complexity (theoretically guided) should be compared and the proper model comparison statistics should be interpreted and reported (e.g., DIC).

12. What would empirically complement Experiment 2 is exposing participants to different levels of viscosities during the reproduction, in which case the predictions regarding the direction of the effect on temporal reproduction would be the opposite. I understand that the effect of viscosity on the speed of movement could be a concern here but when the task is reproducing an interval, I do not think this is a big issue.

13. Although I appreciate the Bayesian Observer Model (as I indicated at the very beginning of my review), I think this is not empirically well-justified given the lack of an effect on the level of CV. Do authors assume that the uncertainty characteristics of the representation itself and reproduction (motor component) are independent enough (I see that they are not) but even in that case, the problem continues since the motor output would inevitably inherits noise that results from the representational uncertainty. Consequently, I think that the motivation behind this cool analysis, with contemporary relevance to the interval timing literature, should be buttressed with stronger empirical or theoretical justifications (of course, while making sure of not being over-speculative).

14. Authors should specify whether the between-trial variability in the core parameters of the DDM was included in the model (as in extended DDM) or not (as in pure DDM).

15. "Analysis of RT values demonstrated faster RTs with longer perceived duration, consistent with previous reports." Please contextualize this finding better in relation to earlier findings and explain the presumed reasons behind it. Authors also do not really elaborate on the finding that the RTs were shortest for the middle viscosity. Is it possible that this finding is also a result of a range effect/Bayesian optimization, where the expectancy was highest for the middle viscosity level?

16. "For the CV, a significant interaction between viscosity and movement length was observed [F(1,27)=7.694, p=0.01, 2p=0.222], in which the CV was significantly lower when subjects moved more, but only when the viscosity was zero [t(27)=-2.237, p=0.034, D=1.2] (Figure 3b), and not for any other viscosity (all p>0.05)." Instead of using a median-split method for the movement length here, I recommend the authors to use a linear mixed-effects model to analyze their data.

I tried to keep my comments as concise as possible not to overwhelm the authors of this fine work but I think that the issues I raised are important on many fronts. I tried to guide authors as much as possible on how they can tackle some of my comments but admittedly they are not necessarily the best ways to address them (e.g. using a blocked design vs. assuming that the thresholds can be set during the timing stimulus in my first point).

Reviewer #3:

The authors required participants to move a robotic arm in environments of varying viscosity. They found that when moving the lever in the presence of high viscosity their estimates of durations of auditory events were compressed. They conducted drift diffusion modelling and two different procedure types to indicate that this was likely due to an influence of action on duration perception, rather than decisional processes. I found the studies interesting but I was unsure how this shaped our understanding of underlying mechanisms, which I think needs clarifying.

1. As the authors outline, previous studies have shown that a variety of movement features impact upon duration judgements. The novelty of their study is suggested to be that movement is restricted by environmental factors rather than 'self' modulations. E.g., the finding reflected in the title of "slowing the body slows down time" has already been demonstrated, by my understanding, in four experiments by Yon et al. (2017, JEPG). However, by my reading both in previous studies and the present studies in fact it is environmental modulations that result in self modulations. I.e., I did not comprehend the difference and this needs clarifying for the theoretical value of these studies to be understood. E.g., previous studies have generated the cells in their design by presenting targets at different locations (e.g., Yon et al., 2017 – external modulation) and participants then choose to move to those targets (volitional component). The present studies create the cells by imposing different forces upon the lever while keeping the targets in the same location. In all studies they are choosing to move to targets at different places and under specific environmental viscosities. The authors describe using sensory feedback due to these environmental features but they would use sensory feedback to monitor both visual target location and forcefields – both of which are constant during a specific movement in previous and present work. Once this line is clarified, what would be the assumed different mechanisms involved in the two cases?

2. In the present studies a number of features of the movement are confounded as viscosity is manipulated. As viscosity increases, the length traversed goes down in a given period of time, the force goes up, the speed goes down. As far as I could see the authors do not disentangle which movement feature was responsible for the influence on duration judgements. My guess is that the correlations are high between these features but perhaps there is enough natural variation in the dataset to disentangle. Could a regression analysis be performed to establish the unique contributions of the different movement parameters? As well as helping the reader to understand what is driving the effects it may resolve some of the inconsistencies with respect to previous literature. E.g., the authors observe that fast movements are associated with increased duration judgements, in contrast with some previous findings. They explain the difference as possibly driven by the fact the conflicting studies have unfilled intervals. However, they do not explain why this may alter the operation of the mechanism. Additionally, there is an experiment in Yon et al. (2017) where high speed is associated with compressed estimates and the intervals are filled here.

3. It would be interesting if the authors could comment upon the precise mechanisms via which each of these features may shape duration perception. Presumably an influence of movement duration on duration perception is easier to explain mechanistically than an influence of distance per se? Why would distance be thought to provide information concerning auditory duration, independently from the distance-duration correlation? E.g., by my understanding size-duration relationships often observed are thought to be mediated by statistical relationships in natural environments, but such that independent information of magnitude would not add to that provided by duration (e.g., Vincent Walsh, TICS, 2003).

---

## [Author Response]

Essential revisions:This is an interesting and well performed study, providing novel notion on the interaction between time perception and reaching movements affected by viscous environments. Nevertheless, we have a set of serious concerns that need to be addressed in the present version of the paper, as follows:1. As the authors outline, previous studies have shown that a variety of movement features impact upon duration judgements. The novelty of their study is suggested to be that movement is restricted by environmental factors rather than 'self' modulations. E.g., the finding reflected in the title of "slowing the body slows down time" has already been demonstrated, by my understanding, in four experiments by Yon et al. (2017, JEPG). However, by my reading both in previous studies and the present studies in fact it is environmental modulations that result in self modulations. I.e., I did not comprehend the difference and this needs clarifying for the theoretical value of these studies to be understood. E.g., previous studies have generated the cells in their design by presenting targets at different locations (e.g., Yon et al., 2017 – external modulation) and participants then choose to move to those targets (volitional component). The present studies create the cells by imposing different forces upon the lever while keeping the targets in the same location. In all studies they are choosing to move to targets at different places and under specific environmental viscosities. The authors describe using sensory feedback due to these environmental features but they would use sensory feedback to monitor both visual target location and forcefields – both of which are constant during a specific movement in previous and present work. Once this line is clarified, what would be the assumed different mechanisms involved in the two cases?

We thank the reviewers for raising the importance of the contrasts between our study and previous studies. We note that indeed, motor control has many intertwined processes of self-modulation that often occur in response to sensory feedback, as shown in the experiments by Yon and colleagues. Our experiments certainty showed that we slowed down movement via viscosity, and that participants displayed a compensatory increase in force. The goal of the Yon experiments was different in that they investigated the association between movement durations and time perception, as compared to movement distance as we did here. They applied manipulations of movement durations during timing first through explicit instruction by asking participants to make finger movements to match a specified criterion of “short” or “long” (and to control for linguistic biasing, “slow” and “fast” in a separate control experiment). In another experiment, they manipulated movement duration by asking participants to manually reach towards a target with a variable distance (5–8 cm for “near” and 15–18 cm for “far”). This was done based on the knowledge that movements are typically longer when they must cover larger distances, and in this case movement duration was implicitly modulated. Notably, these experiments relied on volitional modulation to successfully complete the task (e.g., meeting training criteria for “short” and “long” movements in the first experiment then reaching successfully to “near” and “far” targets). Task success relied on the intentional scaling of movements to fall within a particular range of movement duration or magnitude (as a proxy for movement duration). They found a strong association between perceived duration and movement duration. Our study is distinct from these findings in that movements were scaled exclusively in the spatial, but not temporal domain as a function of viscosity (i.e., movement distance, but not movement duration, scaled with viscosity). This spatial scaling occurred with no explicit instruction or mention of the perturbation, and in contrast to the Yon experiments, the required response was held constant (choice selection in Experiment 1 and temporal reproduction in Experiment 2). Regarding the role of feedback, sensory information in Yon et al. (particularly the visual information in the reaching task) was used to plan movements, but the manipulated sensory information in our experiments (introduced via the somatosensory system) offered no explicit benefit or detriment to completing the task. Participants certainly reacted to these external perturbations by increasing the force they applied, but self-modulation and monitoring was not a critical part of the task as in Yon et al. (2017). We believe this is a critical insight given by our study; even though participants knew that task demands did not change across conditions, durations were still biased by the environmental perturbation – presumably with no conscious awareness of the temporal shift induced. Taken together, these findings point to potentially distinct mechanisms of temporal biasing by movement distance versus duration. In the revised manuscript, we now include a more detailed account of how our study differs from previous work, especially from Yon et al. (2017).

2. The values on bisection point show enormous individual differences between subjects, with a range between 1.5 to 3 seconds, why? In contrast, the effect of viscosity on the BP is very subtle and a large proportion of subject showed negative slope of BP as a function of viscosity (please report). This means that in a subgroup of subjects, an increase in viscosity was accompanied with an increase instead of a decrease in the length of perceived time. This finding does not support one of the mayor conclusions of the paper.

We thank the reviewer for this comment regarding the variability of bisection points. We had wanted to be transparent in our findings and so plotted individual data points wherever possible. First, regarding the variability of the bisection points, we note that, while large, these are within the same range as our previous study using this same task and setup. Author response image 1 shows the BP values from Experiment 1, averaged across viscosity, and the BP values from our earlier report (Wiener, et al. 2019). As demonstrated, the distributions are quite similar. Indeed, Levene’s test of equality finds no significant differences in the variances [*F*(1,47)=0.86392, *p*=0.357].

To the second point, regarding the variability of the viscosity effect, we agree that some subjects did not show any effect of viscosity, and indeed some subjects shifted in the opposite direction. However, we note that more subjects did show the effect than did not, and only a small number shifted in the opposite direction. As such, while the effect of viscosity is not universal, and subject to individual differences (as with any effect in the psychological literature) it is robust on average. To demonstrate this latter point, we now present bootstrapped confidence intervals to demonstrate the robustness of the viscosity effect for both experiments.Further, we note that the variability in the viscosity effect in Experiment 1 may be due to individual differences in how subjects performed the task. Specifically, we noticed large variability in the distribution of movement distances, with some subjects moving a lot and some moving very little, overall. Indeed, subjects were never told explicitly that they *had* to keep moving in Experiment 1 (whereas in Experiment 2 they were forced to keep moving or else reset the trial), and so some subjects experienced more of the viscosity than others. To further disentangle these differences, and as suggested by other reviewer comments, we explored the role of movement distances in mediating the effect. Here, we found that the degree of movement engaged by subjects predicts the viscosity effect; subjects who moved more – and so experienced more viscosity – were also more likely to exhibit a slowing down of time. However, we note that this does not appear to be a continual effect, but rather a stepwise one; that is, the only requirement for the effect to be present is for subjects to be moving, but it is not modulated between-subject by how much one subject moved versus another. Indeed, in Experiment 2, where subjects were required to move the cursor at all times, there is no between-subject correlation. These findings are now described in the text and provided as a figure supplement.

3. In the present studies a number of features of the movement are confounded as viscosity is manipulated. As viscosity increases, the length traversed goes down in a given period of time, the force goes up, the speed goes down. As far as I could see the authors do not disentangle which movement feature was responsible for the influence on duration judgements. My guess is that the correlations are high between these features but perhaps there is enough natural variation in the dataset to disentangle. Could a regression analysis be performed to establish the unique contributions of the different movement parameters? As well as helping the reader to understand what is driving the effects it may resolve some of the inconsistencies with respect to previous literature. E.g., the authors observe that fast movements are associated with increased duration judgements, in contrast with some previous findings. They explain the difference as possibly driven by the fact the conflicting studies have unfilled intervals. However, they do not explain why this may alter the operation of the mechanism. Additionally, there is an experiment in Yon et al. (2017) where high speed is associated with compressed estimates and the intervals are filled here.

We thank the reviewers for this point. Certainly, our dataset is “rich” with variables that can be examined, including movement length, speed, and force. As noted, many of these variables are correlated and also vary with levels of viscosity (as viscosity increases, movement length decreases and force increases). However, given the high correlations, we do not believe it possible to identify a single factor that can explain the findings. Indeed, while our new analysis of movement parameters notes that movement lengths are slightly more correlated with the effect of viscosity than force, neither are a strongly reliable metric at predicting the effect. We therefore suggest that further experiments that specifically manipulate these variables will be necessary to identify which specific factor was most important.

4. Based on DDM fits authors claim that the effect of viscosity was perceptual (and not through decision-biases). This claim is based on the observation that the viscosity level did not affect the starting point or threshold setting but it exclusively modulated the drift rate. Although on the face value, this appears as a valid conclusion based on the model fit results, the methodological details actually put this conclusion into question by invalidating the model fits (unless participants could predict the next viscosity level – see alternative explanations below). Let me elaborate more on this. The standard assumption in the conventional DDM that authors also used in their analysis is that the threshold is set adaptively in response to the context the participant is in (e.g., taking into account the response-to-stimulus interval, reward/penalty ratio, whether instructions emphasize speed or accuracy, prior probability of different alternatives). In other words, the decision threshold and starting point of the decision process are modulated in response to some average representation of the task statistics or in response to a discriminative stimulus that allows participants to predict the characteristics of the next trial (e.g., a trial with vs. without penalty for errors or trial in which the viscosity will be higher vs. lower). But in the current study, the viscosity levels were manipulated randomly from one trial to the next (without any discriminative signaling) and thus participants could not have adjusted the decision thresholds or starting points (unless authors are willing to step out of the conventional DDM domain in which case they have to assume within-trial dynamic modulation of these parameters, which is rather debatable – see Karsilar et al. (2014). Speed Accuracy Trade-off Under Response Deadlines. Frontiers in Neuroscience, 8:248. doi3: 10.3389/fnins.2014.00248). Thus, for DDM to inform us regarding the source of the observed effects, one needs a blocked design where the viscosity levels are kept constant within a given block. What the current results tell me is the following; Yes, the effect of movement on temporal judgments can be mediated by the perceptual effects. What they do not tell me is the following; Movement does not affect the other components of the decision process (for which authors need a different experimental design). I recommend authors to look at Bogacz et al. (2006). The physics of optimal decision making: a formal analysis of models of performance in two-alternative forced-choice tasks. Psychological Review, 113(4), 700-765. https://doi.org/10.1037/0033-295X.113.4.700 for relevant discussions (look at the sections on the optimal performance).

We thank the reviewer for these recommendations, and have read and cited the suggested references. A blocked design would indeed allow us to dissociate viscosity effects to a degree. However, we would predict in this case, where the viscosity is known ahead of time, to exhibit no effect on the perceived duration. Specifically, it is the unpredictable nature of the experienced viscosities, and their trial-to-trial variability that leads to the effect, as the perception of time here is relative to previously experienced intervals. If the intervals were all a single viscosity, the relative differences between intervals would become normalized, resulting in no difference between the conditions. While we would have liked to conduct this additional experiment, we are prevented from doing so due to the present global pandemic.

But there is one way out based on the sequential nature of the diffusion process in the temporal bisection task. At least as it is formulated in Balcı and Simen (2014), which authors cite in the paper, the decision process that would be captured by conventional DDM fits is the second stage diffusion process the parameters of which are contingent on the first stage diffusion (opponent Poisson DDM) process. Within this framework, authors can argue that the viscosity experienced during the movement during the timing of the test stimulus (prior to the offset of the stimulus) can actually modulate the relevant parameters of the second stage diffusion before its launching. If authors are willing to take this route, then they should be mechanistically savvier about such contingencies; otherwise, the results and assumptions are not tied to a concrete process and thus lack the necessary theoretic grounding (as they currently are in the paper). I recommend authors to let the model tie things together, which will bolster the theoretical cohesion of the work.

We thank the reviewers for this and other comments related to the DDM modeling. In response to these concerns, we have substantially expanded our modeling efforts.

For the DDM, we agree with the reviewer’s points regarding the basic DDM and its extensions. In our original planning for this study, we had intended to use the Balci and Simen (2013) model as the basis for interpreting our findings, as we had done in previous work (see Wiener, et al. 2018 and Bader, et al. 2019 in which we used this model). However, to do this would require us to model both an effect of duration and viscosity, as the Balci and Simen model makes specific predictions about changes in model parameters across different durations (as described by the reviewer); unfortunately, as noted in the methods, we lacked a sufficient number of trials to fit the full model using the HDDM method. As we also stated, we attempted to fit the full model, but found large chain instability and large errors. This led to us adopting the “reduced” model in which duration was not explicitly included. Yet, as the reviewer states, this does not preclude us from interpreting our findings through the lens of the Balci and Simen model. Indeed, the accumulation process we display in Figure 4 can be thought of as occurring during the second-stage portion of the Balci and Simen model, but collapsed across intervals. In this way, the changes experienced during the interval by different viscosities can set the parameters for the second-stage process, including potential changes in the threshold, thus obviating concerns about dynamic changes during the interval, such as collapsing threshold boundaries.

Yet, we wished to further demonstrate that our findings would fit within the Balci and Simen model framework. To do this, we again fitted the HDDM model to choice data, but with a separate model that only included duration as a factor, and not viscosity. This model replicates the general findings of Balci and Simen, such as a linear shift in drift rate and starting point, that are both correlated, as well as a change in threshold values close to the bisection point (now included as a figure supplement). We additionally now provide posterior predictive checks for both models to demonstrate that they can provide adequate descriptions of both choice and reaction time data from subjects.

Finally, to further apply this model to our viscosity findings, we used the duration-only HDDM model to generate new data using the viscosity-only HDDM model parameters as weights. To explain further: we generated new data from the duration-only model four times (for each of the four viscosities), by sampling from the posterior distributions generated by HDDM for the duration-only model, then perturbing those parameters for each of the durations with the parameters for each of the viscosities. For example, for drift rate, we took the 7 drift rate values (for each of the 7 durations) and combined each of them with the drift rate for each of the 4 viscosities, which were used to generate 4 datasets in the same manner as the other posterior predictive checks (generating 500 data sets). We observed that the simulated data using these parameters recapitulated the effect of viscosity on the psychometric functions. Thus, while the models were fit separately to duration and viscosity, we were able to combine them to simulate the main findings. We present this here as additional evidence for the suitability of the model, and also now interpret our findings in the context of Balci and Simen, by suggesting that viscosity alters the perception of temporal intervals and so changes the parameters for the second stage process.

I have concerns regarding the use of latent parameter estimates from the HDDM (or slopes estimates gathered from their regression on a factor) in the correlation analysis. As the authors state in the Methods section, HDDM uses the mean overall parameters to constrain the estimates for individual subjects; this is a great analytical tool to increase the statistical power for between-group comparisons but at the same time it makes the estimates for individual subjects unsuitable for correlation analysis. If possible given the number of data points, authors should fit the individual data for estimates to be used in the correlation analyses.

We agree with the reviewer’s concerns regarding “shrinkage” that can occur when hierarchical models are employed (Gelman, 2013). However, we note that the concerns regarding a correlation between point estimates of model parameters for individual subjects and a psychological trait have been discussed elsewhere (Katahira, 2016) and demonstrated in practice (Urai, et al. 2019), suggesting they do not pose an issue. Nevertheless, to address this further, we fit individual subject data using the more traditional Ratcliff method and compared our findings to the hierarchical model. We observed good correspondence between hierarchical and non-hierarchical values for all parameters (see Figure 4—figure supplement 1A). Further, we found that the correlations between model parameters and the viscosity effect also remained the same. Given the similarity of these findings, and other work suggesting using hierarchical estimates is appropriate, we propose to keep the hierarchical findings in the main results, while also noting now that we have confirmed the effects using the non-hierarchical method, presented as a figure supplement.

I have another concern regarding the theoretical handling of the application of DDM to the data. Using DDM in the temporal bisection task actually requires assuming a sequential diffusion process (see Comment 1). This has been previously formulated in Balci and Simen, (2014). Fitting conventional DDM is just an approximate method to capture the behavioral output of the sequential process. This sequential process requires yoking the drift-rate and starting point during model fits or testing the relationship between them to validate the model in the context of the dataset at hand. Based on this model, if one assumes that viscosity of the movement affects the perceptual/timing process (as hypothesized in this work) then one would not only expect a relationship between viscosity and the drift rate but also a relation between viscosity and the starting point. This issue should be addressed in the paper.

We thank the reviewer for noting this. As we described above, the drift rate and starting point should indeed be correlated, but this requires duration to be a factor in the model. In our original presentation of results, duration was not a factor, and so one would not, necessarily, suppose that drift rate and starting point need to be correlated. In our expanded modeling efforts, we now show that this is the case, with starting point and drift rate correlated across durations. However, the reviewer is also correct in stating that a change in the starting point or drift, as interpreted by the Balci and Simen model, could be construed as affecting the perceptual stage. We now include this in our manuscript.

5. The quality of DDM fits are not shown in the paper. The quality of the fits should be certainly demonstrated for both models. Please provide an example of the actual and predicted DDM psychometric and chronometric functions.

As described above, we now provide posterior predictive checks for the DDMs in the paper. In addition, we include deviance information criterion (DIC) scores for both models. Notably, the result of this comparison suggests that, while the starting point is necessary for the duration-only DDM, it is not warranted for the viscosity-only DDM, suggesting the starting point is not needed to describe the effect.

We additionally provide model fits to the Bayesian Observer Model, as part of our expanded efforts (see below).

6. The fact that the model parameters (DDM and Bayesian) show a non-linear relation with viscosity does not imply that they could also have a role in explaining the behavior. A downstream non-linear reader could produce the observed behavioral effects of viscosity. Since the changes in DDM threshold and bias as a function of viscosity are nonlinear, it would be interesting to correlate the individual slope of BP x viscosity with the negative coefficient on an exponential or power function fitted to these parameters vs viscosity.

We thank the reviewer for raising this point. We agree that there are non-linear relations with viscosity in both models. We followed the reviewer’s suggestion of using the negative coefficient from a power function instead of the slope from a linear one. However, the results remained the same for both the DDM and Bayesian model and the parameter relations with behavior.

7. The authors suggest that the DDM results support the notion that the shift in BP produced by larger viscosities is due to a decrease in drift rate, which in turn produced changes in the perceived duration rather than decision parameters. However, the authors also suggest that time perception and decision making are intermingled in the bisection task and that once the categorical boundary has been crossed, subjects stop accumulating temporal information (Wiener and Thompson, 2015). How do you reconcile the two results?

We thank the reviewer for raising this. As described above, we have expanded our modeling and now interpret our findings according to the Balci and Simen model. In this case, the DDM is explicitly modeling the second-stage process, which occurs after interval offset. While this is now discussed in the manuscript, we additionally note that the reproduction task does not share the same issues regarding the categorical boundary.

8. Although I appreciate the Bayesian Observer Model (as I indicated at the very beginning of my review), I think this is not empirically well-justified given the lack of an effect on the level of CV. Do authors assume that the uncertainty characteristics of the representation itself and reproduction (motor component) are independent enough (I see that they are not) but even in that case, the problem continues since the motor output would inevitably inherits noise that results from the representational uncertainty. Consequently, I think that the motivation behind this cool analysis, with contemporary relevance to the interval timing literature, should be buttressed with stronger empirical or theoretical justifications (of course, while making sure of not being over-speculative).

We thank the reviewer for raising these points, as it indicates that we were not clear in justifying the use of the model. To address this, we now provide further details regarding the motivation of the model. First, we note that, even though the CV did not change with viscosity, the slope of reproduced durations did, and it remained possible that this change was occurring due to measurement or production noise, and so we wanted to explicitly model this using the Bayesian model. Second, as with the DDM, we have now expanded the modeling section for Experiment 2, in which 1) we now provide simulated data derived from the model fits demonstrating that the model can reproduce the viscosity effect, and 2) we compare these results to a second, modified version of the Bayesian model. In the modified version, we specifically moved the offset from the measurement stage to the production stage, so as to test if the model specifically requires a change in measurement, or in production, to match the effect of viscosity. For this latter effort, we found that only the original model could reproduce the findings, and further the original model provided a better fit to the experimental data. These results bolster our findings that the effect of viscosity occurs during measurement, and is not introduced later when subjects reproduced the interval (incidentally, we note that viscosity was *only* experienced during measurement, and so one would not necessarily predict an effect during production, but we wanted to test this in our models).

9. It would be interesting if the authors could comment upon the precise mechanisms via which each of these features may shape duration perception. Presumably an influence of movement duration on duration perception is easier to explain mechanistically than an influence of distance per se? Why would distance be thought to provide information concerning auditory duration, independently from the distance-duration correlation? E.g., by my understanding size-duration relationships often observed are thought to be mediated by statistical relationships in natural environments, but such that independent information of magnitude would not add to that provided by duration (e.g., Vincent Walsh, TICS, 2003).

We thank the reviewers for this suggestion to expand on the mechanism responsible for the compressing effect of viscosity on time estimates. We first note that movement duration in our study design was not set up to vary as a function of viscosity. Rather, movement durations adhered closely to the duration of auditory tones, and the magnitude of movements decreased with viscosity. The distance-duration correlation is certainly relevant to the sensorimotor time estimate, but the decrease in auditory time as a function of viscosity indicates a cross-modal interaction. Our results suggest that the two channels of temporal processing were not redundant; if this had been the case, there would have been no effect of movement on the separate auditory estimate. We propose that these effects can be approached from a perspective of optimal cue combination. In a previous study (Wiener et al. 2019), we found that movement (versus no movement) enhanced temporal precision. As discussed in the manuscript, movement can also bias timing (e.g., Yon et al. 2017; Yokosaka et al. 2015, Press et al., 2014), and from a cue combination perspective (Ball, et al. 2017), we suggest that auditory timing in our experiments reaped a benefit from integrating motor information with higher precision, at the cost of a slight bias. Our findings are compatible with Walsh’s Theory of Magnitude (ATOM; Walsh, 2003) with respect to the common neural mechanisms of spatial and temporal processing. More specifically, our experiments provide new evidence to a larger body of literature that modulating the spatial characteristics of movements can modify their perceived durations. A consideration not included in the original theory is the role of simultaneous temporal measurements from different sensory channels, and this gap can potentially be approached with principles of cue combination – for example, how does the brain unify temporal estimates from noisy sensory pathways that are susceptible to modality-specific biases? In the updated Discussion section we have outlined these considerations, with a particular focus on cue combination and ATOM.

10. Authors talk about the self-regulation of movement vs. induced constraints on the movement and at times they allude to the embodied cognition. This brings a relevant issue to my mind; within the semi-theoretical stance taken in the paper what would the predictions be for the effect of observed biological vs. non-biological movement on time perception and how do such predictions compare to the results reported in the literature? This could be one way in which the paper can set a stronger theoretical foundation and consolidate some of the other findings in the literature under an overarching theoretical framework in light of the current findings reported in the paper. I see this as a great opportunity for the authors/paper.

We thank the reviewers for highlighting the importance of biological motion in temporal perception. Visual timing experiments in this domain demonstrate interesting effects: when timing the duration of a dot moving across a screen with different movement profiles, biological motion (compared to constant motion) is timed more precisely for sub- and supra-second intervals, and is also more accurate for sub-second intervals (Gavazzi et al. 2012). Wang and Jiang (2012) found that when participants viewed a light-point display of a human walking, the duration of the display was perceived as longer than a static display or a display with non-biological motion. Importantly, the duration of biological motion was still expanded when the dot positions were scrambled (i.e., no longer in the shape of a person).

There are fewer studies available that investigated the role of generated (rather than simply observed) biological motion, but this is an interesting question to examine. Carlini and French (2014) previously were able to manipulate generated movement profiles via a cursor tracking task. Participants timed the duration of a target (visually or both visually and manually) moving across a screen. Hand tracking overall improved both accuracy and precision, but a close examination of the different types of hand tracking revealed notable dissociations; when the dot exhibited biological motion (smooth acceleration, peak, and deceleration), the duration was perceived as shorter than targets with non-biological movement profiles (i.e.., either a constant velocity or a sharp “triangular” profile). The experimenters also manipulated the velocity (degrees/second) and found that the underestimation of biological motion occurred only at the lowest velocity of 5°/second – at all other velocities, biological motion was timed most veridically. Broadly, these results are consistent with our findings. If we were to test a similar task with a viscosity manipulation, we predict that manual tracking of biological movement would be more accurately timed, similarly to Carlini and French (2014). This would also allow us to examine the separate contributions of force and movement distance; to examine movement distance we would scale the cursor speed down with viscosity (thereby requiring the same amount of cumulative force between viscosities), and to examine force we would keep the speed constant (requiring a compensatory force increase with viscosity).

Another way to assess timing of biological motion would be to test passive movement. The robotic arm would move participants with a pre-selected movement plan, and this would remove the potentially confounding effect of visual cueing from the cursor. Predictions for this paradigm are slightly more challenging because passive movement during timing is not well-explored in the literature; however, this experiment would allow for full control of movement variables in a way that is not usually available. In the updated manuscript we include a brief discussion on biological motion and some of the variables to consider with experiments of this nature.